# ACT: AGENTIC CLASSIFICATION TREE

## ABSTRACT

When used in high-stakes settings, AI systems are expected to produce decisions that are transparent, interpretable, and auditable—a requirement increasingly expected by regulations. Decision trees such as CART provide clear and verifiable rules, but they are restricted to structured tabular data and cannot operate directly on unstructured inputs such as text. In practice, large language models (LLMs) are widely used for such data, yet prompting strategies such as chain-of-thought or prompt optimization still rely on free-form reasoning, limiting their ability to ensure trustworthy behaviors. We present the *Agentic Classification Tree* (ACT), which extends decision-tree methodology to unstructured inputs by formulating each split as a natural-language question, refined through impurity-based evaluation and LLM feedback via TextGrad. Experiments on text benchmarks show that ACT matches or surpasses prompting-based baselines while producing transparent and interpretable decision paths.

---

**Unstructured Data: Tuberculosis diagnosis based on symptoms descriptions**

**Patient 1:** "My main symptom is a runny nose with increased sweating, pain, fever, skin rashes, nasal congestion, sore throat, muscle pain, loss of appetite, and chills. The heavy pain is in my left temple, and I have pink rashes on the back of my neck." 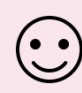

**Patient 2:** "I am coughing blood with pain (heavy in my left temple), skin lesions, nasal congestion, extreme fatigue affecting activities, diffuse muscle pain, loss of appetite, chills, a heavy pain on my left temple, and pink rash on the right side of my neck." 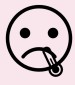

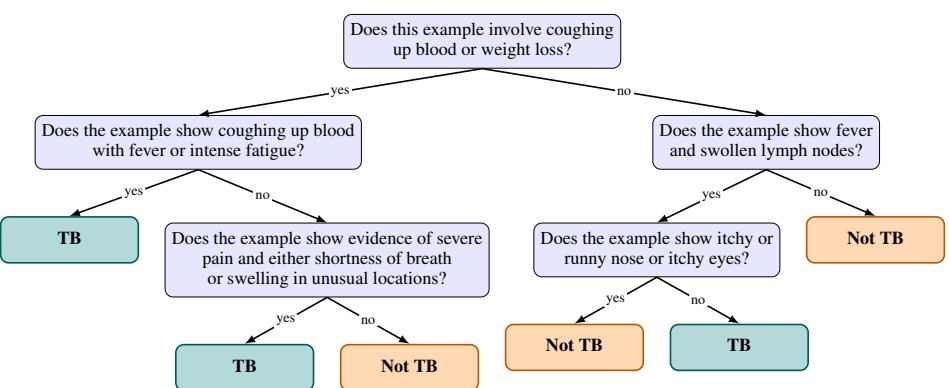

Figure 1: Example ACT decision tree for tuberculosis diagnosis using unstructured, free-text patient descriptions. A tree is automatically learned, with each node containing a binary natural language question, autonomously discovered via recursive prompt refinement to maximize label separation at each split. At inference, these questions are answered by a large language model (LLM) from the root node to the leaves of the tree. The final classification (TB or Not TB) corresponds to the majority label of training examples described by each leaf.

## 1 INTRODUCTION

As AI systems become increasingly integrated into high-stakes domains such as healthcare, education, legal decision-making, and finance, the need for transparency, interpretability, and auditability in AI decision-making has intensified. These requirements are grounded not only in practical considerations, but also in legal obligations: recent governance frameworks and regulations such as the EU AI Act[1], the OECD AI Principles[2], and the NIST AI Risk Management Framework[3] emphasize that AI-based decisions in high-stakes scenarios must be explainable and subject to human oversight. In this context, there is a growing need for models whose decision processes can be inspected, verified, and understood—not only by technical experts but also by stakeholders, auditors, and regulators.

Historically, complex decision-making tasks in high-stakes domains—such as medical diagnosis or fraud detection —have been addressed on interpretable AI systems such as expert systems (Jackson, 1986; Shortliffe, 1986; Leonard, 1993; Talebzadeh et al., 1995) and decision trees (Breiman et al., 1984). These models allow users to trace each decision through a sequence of explicit, human-understandable rules Slack et al. (2019), directly supporting the requirements set out by regulatory frameworks[4]. However, their applicability is largely limited to structured inputs such as tabular data, and are not directly applicable to unstructured data such as natural language or images.

Recent advancements in large language models (LLMs) have substantially enhanced the capacity for semantic understanding and reasoning within natural language processing (Brown et al., 2020; Achiam et al., 2023). Building on these models, agentic systems (Shinn et al., 2023; Yao et al., 2023; Schick et al., 2023) have been proposed to address complex tasks involving reasoning over diverse types of data, and are seen as a very promising direction for future AI systems (Chen et al., 2023; Wang et al., 2024).

Yet, despite their impressive capabilities, LLMs still suffer from several drawbacks—such as hallucinations, inconsistencies, unreliability, and difficulty to audit (Kryściński et al., 2019; Ji et al., 2023a; Tamkin et al., 2021). These issues constitute major obstacles to building reliable AI systems, limiting their adoption in sensitive or regulated domains. A growing body of work seeks to address these weaknesses by guiding LLM behavior toward more structured or self-consistent reasoning. Chain-of-Thought prompting Wei et al. (2022), for example, encourages multi-step reasoning by inserting explicit intermediate steps into prompts, improving consistency and sometimes reducing hallucinations. Other techniques—such as self-reflection Shinn et al. (2023); Madaan et al. (2023) and prompt optimization Zhou et al. (2022); Ji et al. (2023b); Yuksekgonul et al. (2024); Renze & Guven (2024)—introduce agentic feedback loops to iteratively refine model outputs. While these methods improve reliability in practice, they still rely on free-form text generation and implicit reasoning patterns that remain difficult to verify, audit, or formally constrain.

In this paper, we introduce a new type of classifier designed to combine the semantic reasoning capabilities of large language models (LLMs) with the transparency of decision trees. The idea is to follow a divide-and-conquer paradigm, allowing to reduce hallucination and reasoning errors by successive decomposition of the problem following a hierarchical decision logic.

We therefore propose to structure LLM and agentic reasoning to address complex classification tasks through verifiable decision patterns on *unstructured data* such as images or texts, in the form of an agentic decision tree. Adapting the traditional decision tree algorithms such as CART Breiman et al. (1984) and C4.5 Salzberg (1994), we propose ACT, an agentic decision tree where each node is defined as a binary natural language question over the input data, iteratively partitioning the data into two subsets at each step. The best splits are found using TextGrad Yuksekgonul et al. (2024), a prompt-refinement technique providing textual feedback to an agent, to minimize the Gini criterion Breiman et al. (1984) at each split. Ultimately, ACT takes the form of a decision tree (cf. Figure 1), each leaf leading to a class prediction after successive splitting.

The benefits of ACT are therefore the following:

---

[1]https://artificialintelligenceact.eu/

[2]https://oecd.ai/en/ai-principles

[3]https://www.nist.gov/itl/ai-risk-management-framework

[4]AI RMF Playbook, https://airc.nist.gov/docs/AI_RMF_Playbook.pdf

- **Transparent and structured decision process:** ACT enforces a systematic, rule-based structure that aligns the model's internal reasoning with its observable decision process. Its tree-based architecture provides fully traceable and interpretable decision paths, facilitating auditability, human oversight, and intervention—akin to the properties of traditional expert systems.

- **Optimized decision process:** we show that decomposing some tasks into subquestions improves the performance of LLMs without retraining. By automatically finding the most relevant questions at each node, ACT is able to surpass existing methods.

## 2 PROBLEM SETUP AND RELATED WORK

We consider a binary classification problem over an *unstructured* input space $\mathcal{X}$ (e.g., text or images), with labels $y \in \{0, 1\}$. Given a dataset $\mathcal{D} = \{(x_i, y_i)\}_{i=1}^N$ sampled i.i.d. from an unknown distribution, the goal is to learn a classifier $f : \mathcal{X} \to \{0, 1\}$ that is both accurate and interpretable. There is an abundant literature in XAI that focuses on defining desiderata for interpretable models Doshi-Velez & Kim (2017); Lipton (2018). Building on the regulatory requirements discussed in Section 1, we therefore aim to build a decision model satisfying the following properties:

- **Accuracy:** competitive predictive performance on unstructured inputs;
- **Transparency:** decisions should be produced via interpretable, step-by-step reasoning;
- **Contestability:** the ability to contest an algorithm-based decision by disputing the conditions it relies on Wachter et al. (2017); Venkatasubramanian & Alfano (2020); Lyons et al. (2021).

Translating these desiderata into technical requirements is made especially difficult given the opaqueness and complexity of modern architectures such as LLMs and VLMs. There is a clear lack of literature support for what explanations satisfying these properties should look like in these contexts. As a result, in this work, we propose to define interpretability through the notion of **explicit decision paths**—structured sequences of semantically meaningful reasoning steps that can be inspected, understood, and verified by human users. Below, we ground this proposition in the existing XAI and LLM literatures.

**Interpretable models and post-hoc explanations** Interpretable models such as decision trees (Breiman et al., 1984; Salzberg, 1994) are traditionally viewed as a natural solution to meet the three aforementioned criteria. Offering transparent, rule-based decision paths that are easily understood and audited, their hierarchical structure allows users to trace predictions through explicit, human-readable splits, making them well-suited for regulated or high-stakes domains (Rudin, 2019). However, they are fundamentally limited to structured inputs, relying on predefined features and struggling to handle unstructured data such as raw text or images. Efforts to address interpretability in unstructured data contexts include post-hoc explanation techniques such as LIME (Ribeiro et al., 2016) and SHAP (Lundberg & Lee, 2017) proposing explanations in the form of feature attributions at the pixel (for images) or word (for text) level. Although very popular, these methods have been criticized for often failing to capture the model's true reasoning (Rudin, 2019; Laugel et al., 2019), consequently offering limited guarantees for auditability, consistency, or regulatory compliance.

More recent efforts in neural NLP have attempted to incorporate explanations directly into the model's output, for instance through self-explanation (Nye et al., 2021) or agentic feedback loops (Madaan et al., 2023). However, these approaches rely on free-form natural language generation, which—although interpretable to humans—remains unverifiable and unstructured. As a result, they fall short of providing a consistent and auditable decision procedure.

**Workflow and prompt optimization** Research on building trustworthy LLM and VLM-based systems has largely focused on workflow design and prompt refinement. Approaches such as AutoPrompt (Shin et al., 2020), TextGrad (Yuksekgonul et al., 2024), and DSPy (Khattab et al., 2023) automatically optimize the prompt instructions to better align the LLM behavior with downstream tasks. These methods show that LLMs can be guided toward more accurate or systematic outputs without retraining, but do not provide explanation patterns. Techniques such as Chain-of-Thought prompting (Wei et al., 2022), ReAct (Yao et al., 2023), and Reflexion (Shinn et al., 2023) structure

Table 1: Side-by-side comparison between traditional decision tree algorithms (CART, C4.5) and our proposed Agentic Classification Tree (ACT).

|  | **CART / C4.5** | **Agentic Classification Tree** |
|---|---|---|
| Data type | Tabular | Unstructured |
| Node definition | Numerical: binary threshold (both). Categorical: CART binary subset; C4.5 multiway split. | Binary natural-language question queried to an LLM over the inputs. |
| Best split search | Greedy exhaustive search over candidate splits. | Iterative refinement of the question through prompt optimization (TextGrad) with LLM feedback. |
| Split criterion | Gini (CART) or Information Gain Ratio (C4.5). | (Gini or IG) combined with semantic purity analysis. |
| Inference method | Each input follows the tree based on feature values. | Each input is queried by the LLM at each node and routed according to its answers. |

reasoning through intermediate steps or self-feedback, providing both performance increases and transparent decision patterns. However, their reliance on free-form text generation makes them still vulnerable to hallucinations, questioning their efficacy in auditing and verification contexts.

We propose to address these challenges by enforcing a hierarchical tree-structured decision process over unstructured inputs. Defining each node as a natural-language question optimized through LLM feedback ensures both semantic adaptability and transparent reasoning paths. Our proposed framework, called Agentic Classification Tree (ACT), thus aims to combine the interpretability of decision trees with the semantic reasoning of LLMs, providing an accountable alternative to existing black-box prompt optimization methods.

## 3 METHODOLOGY

To build a structured agentic workflow in the form of a classification tree, we take inspiration from traditional decision tree algorithms such as CART (Breiman et al., 1984) and C4.5 (Salzberg, 1994). Table 1 provides a side-by-side comparison between traditional decision trees and our proposed ACT method.

### 3.1 TRADITIONAL DECISION TREES: CART AND C4.5

Classical decision-tree algorithms such as CART (Breiman et al., 1984) and C4.5 (Salzberg, 1994) recursively partition a dataset based on feature values. At each node, a split is defined by either a threshold on a numerical feature or a subset of categorical values. The quality of each split is measured by an impurity criterion—typically Gini impurity (CART) or Information Gain ratio (C4.5)—and the algorithm greedily selects the split that yields the greatest impurity reduction, continuing recursively until a stopping criterion is met (e.g., maximum depth or node purity).

These methods are efficient and interpretable (Molnar et al., 2020) for structured, tabular data, but cannot operate directly on unstructured inputs such as text or images. This limitation motivates our proposed ACT, which replaces feature-based splits with optimized natural language queries.

### 3.2 PROPOSED APPROACH: AGENTIC CLASSIFICATION TREE (ACT)

The Agentic Classification Tree (ACT) is designed to extend classical decision tree methods to unstructured data. Like CART and C4.5, ACT recursively partitions the input space, but instead of relying on numeric or categorical features, it leverages large language models (LLMs) to define natural-language queries that guide data splits.

**Node definition** Each node $x$ in ACT is defined by a prompt $p$ to semantically partition data via an LLM. Given an instance $x_i$, the LLM is queried with $p$ and responds with a binary answer (`yes` or `no`), determining the branch assignment. This process defines the following split:

$$\mathcal{D}_L^x = \{(x_i, y_i) \in \mathcal{D}^x \mid f_{\text{split}}(p, x_i) = \text{yes}\}, \quad \mathcal{D}_R^x = \mathcal{D}^x \setminus \mathcal{D}_L^x. \tag{1}$$

where $f_{\text{split}}(p, x_i)$ denotes the LLM's response to prompt $p$ on input instance $x_i$. The resulting subsets $\mathcal{D}_L^x$ and $\mathcal{D}_R^x$ thus reflect the semantic partition induced by the model at node $x$.

To initialize the splitting process, we begin with a neutral, generic question such as:

> `"Based on the provided example, does it belong to the positive class? (yes/no)"`

This provides an unbiased starting point. Since $f_{\text{split}}$ is neither trained on the task nor informed of the target labels, the initial partition is not expected to be meaningful. This is by design: withholding task-specific information ensures that decision prompts are derived from the training data rather than from prior knowledge. In contrast to approaches such as TextGrad (Yuksekgonul et al., 2024), which typically begin with a task-specific question and iteratively refine it, ACT starts from a neutral prompt and discovers meaningful questions from scratch through data-driven refinement.

**Best split criterion** ACT optimizes each decision node by iteratively refining a natural-language question $p^{(k)}$ to achieve an effective semantic partition of the data. To guide this process, it combines quantitative evaluation (e.g., Gini impurity) with qualitative feedback from the LLM, yielding both statistical and semantic insights for prompt refinement. The procedure consists of two components:

1. **Quantitative Impurity Evaluation.** As in classical decision tree algorithms such as CART and C4.5, the quality of a split induced by the current prompt $p^{(k)}$ is evaluated using standard impurity criteria (e.g., weighted Gini impurity (Breiman et al., 1984) or information gain ratio (Salzberg, 1994)). We denote this score by $\delta(p^{(k)})$, which serves as an objective function quantifying how well the semantic split separates the class labels.

2. **Semantic Purity Analysis via LLM.** While the quantitative evaluation (e.g., Gini impurity) measures class-label heterogeneity within each partition, it does not reveal the *semantic* factors underlying the impurity. To make these sources explicit, we analyze each child node independently by contrasting its class-conditional subsets (correct vs. misclassified examples). For this purpose, we align the predicted answer (`yes`/`no`) with the ground-truth label ($y_i \in \{0, 1\}$) when defining correct and incorrect routing. Although this introduces an artificial correspondence between branch outcomes and class labels, it proved to be more effective in practice: the LLM more readily exploits feedback framed in terms of correct versus incorrect predictions than feedback based on contrasting the two branches directly (e.g., retaining the majority class and pruning the minority one). This alignment does not prevent the emergence of negative queries (e.g., "*Is feature A absent?*"), which the LLM can in principle generate and refine, though such formulations are typically harder to elicit consistently.

   Formally, each subset $g \in \{\mathcal{D}_L^x, \mathcal{D}_R^x\}$ is associated with the model's predicted answer $\hat{y}_g$ to the current prompt. For convenience, we denote by $\hat{y}_g \in \{0, 1\}$ the numeric encoding of this answer ($\hat{y}_g = 1$ for "yes", $\hat{y}_g = 0$ for "no"). We then further partition each subset into two groups:

$$X_{\text{correct}}^x = \{(x_i, y_i) \in g \mid y_i = \hat{y}_g\}, \quad X_{\text{error}}^x = \{(x_i, y_i) \in g \mid y_i \neq \hat{y}_g\}.$$

   Specifically, $X_{\text{correct}}$ consists of input–label pairs where the LLM's predicted label $\hat{y}_g$ matches the true label $y_i$, while $X_{\text{error}}$ consists of misclassified pairs where $y_i \neq \hat{y}_g$.

   Next, we prompt the LLM to analyze these two groups and identify key semantic **characteristics** that distinguish them—i.e., features whose presence or absence could explain why some examples are misclassified. Based on this contrast, the LLM—denoted $f_{\text{purity}}$—returns concise, actionable feedback $s_g$ that is then used to refine the node's splitting question in the next optimization step to reduce impurity. By performing this semantic analysis independently on subsets $D_L$ and $D_R$, we obtain targeted feedback $s_{D_L}$ and $s_{D_R}$ that are then used to guide subsequent question-refinement iterations.

**LLM Task: Semantic Purity Feedback for "yes" group** ($D_L$)
Below are two groups of samples for which a model answered either "yes" or "no" in response to a natural language prompt to predict their class label.
Provide feedback on key **characteristics** that are present or absent in the group where the model's predicted label matched the true label, and in the group where the prediction was incorrect.
For the following examples, the model answered "yes":

- **Well-classified examples** (true label = "yes"):
  *[List of correctly classified inputs]*
- **Misclassified examples** (true label = "no"):
  *[List of misclassified inputs]*

The feedback you provide must be clear and concise. Focus on the one or two most important **characteristics**.

**Best split search**  At each decision node, the objective is to refine the natural-language prompt $p^{(k)}$ so that the resulting partition minimizes the weighted Gini impurity $\delta(p^{(k)})$. As described previously, this quantitative criterion is complemented by semantic feedback $(s_{D_L}, s_{D_R})$, obtained from LLM-based analyses of misclassified versus correctly classified examples. To jointly evaluate statistical impurity and semantic error patterns, we define a semantic–impurity objective:

$$\mathcal{L}(p^{(k)}) = f_{\text{loss}}(p^{(k)}, s_{D_L}, s_{D_R}, \delta(p^{(k)})),$$

where $f_{\text{loss}}$ is instantiated as a large language model (LLM). Given the current question, impurity statistics, and semantic feedback, the LLM returns a natural-language diagnosis of its limitations, highlighting semantic factors that contribute to impurity.

We realize the refinement process using the differentiable prompting framework *TextGrad* (Yuksekgonul et al., 2024). Each refinement iteration proceeds in two stages:

(i) *Guidance stage:* `TextGrad.feedback` takes as input the prompt $p^{(k)}$ together with its evaluation $\mathcal{L}(p^{(k)})$, and generates a natural-language editing instruction $\nabla_p \mathcal{L}(p^{(k)})$ that specifies how the prompt should be revised.

(ii) *Revision stage:* `TextGrad.step` applies this instruction to produce an updated prompt:

$$p^{(k+1)} = \texttt{TextGrad.step}\Big(p^{(k)}, \nabla_p \mathcal{L}(p^{(k)})\Big).$$

The optimization loop is repeated for up to $K$ steps or until convergence. At each iteration, the current prompt $p^{(k)}$ is used to partition the data, evaluate impurity, and analyze semantic error patterns via the LLM, as described above. These signals jointly inform the loss $\mathcal{L}(p^{(k)})$, which is then used by `TextGrad` to generate a refined prompt $p^{(k+1)}$. Among all prompts generated during the optimization process, $p^{(0)}, p^{(1)}, \ldots, p^{(K)}$, we select the one with the lowest impurity on the training set, i.e., $p^* = \arg\min_k \delta(p^{(k)})$.

**Tree construction**  ACT builds the decision tree recursively in a top-down manner. At each node, the best prompt is optimized via the procedure described above. If the resulting subset is pure or meets a predefined stopping criterion (e.g., minimum node size — the number of training examples in the node, Gini impurity below a threshold, or maximum depth), a leaf node is created. As in classical decision trees, the leaf is assigned the majority class label among the examples it contains. The complete ACT procedure is summarized in Algorithm 1.

**Tree interpretability**  Decision trees, which rely on transparent structures and parameters to make decisions, are widely considered to be interpretable Molnar et al. (2020). Owing to its shallow decision-tree structure and natural-language node questions, ACT exhibits the same desirable properties. In practice, just like general decision trees, interpretability depends on both the tree depth (i.e., the number of questions along a decision path) and the clarity of the questions. For this reason, we limit the tree depth to three or four levels to keep decision paths concise, and we explicitly instruct `TextGrad` during training to constrain generated questions to at most two logical clauses (AND/OR), as detailed in Appendix A.2. This generally tends to keep the resulting questions easy to understand.

---

**Algorithm 1:** ACT: Agentic Classification Tree

---

**Input:** Dataset $D$
**Output:** Decision tree $T$
$T \leftarrow \texttt{GrowPromptTree}(D)$ ;
**return** $T$ ;

**Function** $\texttt{GrowPromptTree}(D)$:
    Initialize prompt $p^{(0)}$ ;                          `// default initialization`
    **if** *D is pure or stopping criterion is met* **then**
        **return Leaf** node with majority class label;
    **for** $k = 0$ **to** $K - 1$ **do**
        $D_L \leftarrow \{(x_i, y_i) \in D \mid f_{\text{split}}(p^{(k)}, x_i) = \texttt{yes}\}$;
        $D_R \leftarrow D \setminus D_L$;
        $\delta(p^{(k)}) \leftarrow \frac{|D_L|}{|D|} \cdot \text{Gini}(D_L) + \frac{|D_R|}{|D|} \cdot \text{Gini}(D_R)$ ;
        **foreach** $(g, \hat{y}_g) \in \{(D_L, \texttt{'yes'}), (D_R, \texttt{'no'})\}$ **do**
            $X_{\text{correct}} \leftarrow \{(x_i, y_i) \in g \mid y_i = \hat{y}_g\}$ ;
            $X_{\text{error}} \leftarrow \{(x_i, y_i) \in g \mid y_i \neq \hat{y}_g\}$ ;
            $s_g \leftarrow f_{\text{purity}}(p^{(k)}, X_{\text{correct}}, X_{\text{error}})$ ;  `// Intra-group semantic analysis`
        $\mathcal{L}(p^{(k)}) \leftarrow f_{\text{loss}}(p^{(k)}, s_{D_L}, s_{D_R}, \delta(p^{(k)}))$ ; `// LLM-evaluated semantic loss`
        $\nabla_p \mathcal{L}(p^{(k)}) \leftarrow \texttt{TextGrad.feedback}(p^{(k)}, \mathcal{L}(p^{(k)}))$ ;
        $p^{(k+1)} \leftarrow \texttt{TextGrad.step}(p^{(k)}, \nabla_p \mathcal{L}(p^{(k)}))$ ;
    $p^* \leftarrow \arg\min_{p^{(k)}} \delta(p^{(k)})$ ;                   `// Prompt with lowest impurity`
    Split $D$ using $p^*$ into $D_L$ and $D_R$ ;
    node $\leftarrow$ Create prompt node with final question $p^*$ ;
    node.left $\leftarrow \texttt{GrowPromptTree}(D_L)$ ;
    node.right $\leftarrow \texttt{GrowPromptTree}(D_R)$ ;
    **return** node ;

**Function** $\texttt{Gini}(D)$:
    **return** $1 - \sum_k p_k^2$, where $p_k$ is the class proportion in $D$;

---

## 4 EXPERIMENTS

We empirically evaluate the proposed Agentic Classification Tree (ACT) against state-of-the-art baseline methods on multiple binary text classification datasets.

**Datasets.** We evaluate ACT on five binary text-classification tasks spanning medical diagnosis, content moderation, sentiment analysis, and structured customer-attribute prediction. The **DIAGNO** dataset contains short clinical descriptions labeled as tuberculosis (TB) or not TB, and we use balanced train–test splits for controlled evaluation. **SPAM** consists of email messages annotated as spam or ham, also evaluated with balanced splits. **JAILBREAK** contains user prompts labeled according to whether they attempt to circumvent safety policies, and is used in its original moderately imbalanced form. **BANKCHURN** is a customer-churn dataset in which tabular attributes are serialized into short textual profiles following the protocol used by (Arzaghi et al., 2025). This task emphasizes dataset-specific attribute relationships rather than broad world knowledge. **IMDB** is a standard sentiment dataset of movie reviews, from which we use a balanced subset to maintain consistency across tasks.

**Models.** We evaluate our method on four language models: Gemma-4B Team et al. (2025), GPT-Nano-4.1 Achiam et al. (2023), GPT-Mini-4.1 Achiam et al. (2023), and Qwen3-4B Yang et al. (2025). These models were selected to represent diverse architectural approaches while prioritizing computationally efficient smaller models. To assess the effect of model scale, we run all main experiments on the Nano and Mini variants of GPT-4.1 and report additional scaling results with full GPT-4.1 Achiam et al. (2023) in Appendix E.1, together with a detailed cost analysis in Appendix G.

Table 2: Comparison of classification methods across five datasets (DIAGNO, SPAM, JAILBREAK, BANKCHURN, IMDB) and four LLMs. Results show accuracy and F1-score on the test set (in %) for nine benchmarks, as well as the proposed ACT with varying hyperparameters ($d$: tree depth, $k$: optimization steps per node).

| Method | Gemma3 4b Acc | F1 | GPT-4.1 Nano Acc | F1 | GPT-4.1 Mini Acc | F1 | Qwen3 4b Acc | F1 | Avg Acc | F1 |
|---|---|---|---|---|---|---|---|---|---|---|
| **DIAGNO** | | | | | | | | | | |
| CoT (0-shot) | 61.5 | 69.5 | 63.3 | 54.8 | 61.2 | 51.4 | 63.2 | 41.7 | 62.3 | 54.4 |
| DSPy (BFSR, 8 demos) | 64.0 | 68.2 | 68.7 | 61.2 | 64.5 | 59.4 | 58.8 | 54.9 | 64.0 | 60.9 |
| TEXTGRAD | 63.3 | 56.7 | 64.5 | 53.0 | 65.8 | 55.6 | 64.0 | 60.0 | 64.4 | 56.3 |
| TF-IDF + CART ($d$=3) | —— | —— | —— | —— | —— | —— | —— | —— | 78.8 | 78.9 |
| TF-IDF + XGBoost ($d$=3, $nt$=200 ) | —— | —— | —— | —— | —— | —— | —— | —— | 81.7 | 82.0 |
| BERT | —— | —— | —— | —— | —— | —— | —— | —— | 90.3 | 91.1 |
| ROBERTA | —— | —— | —— | —— | —— | —— | —— | —— | 89.8 | 90.4 |
| Con. tag. | 65.5 | 63.9 | 59.3 | 58.1 | 75.3 | 78.6 | 58.8 | 64.7 | 64.7 | 66.3 |
| Rule Fit | | | | | | | | | 80.2 | 79.3 |
| ACT ($d$=3, $k$=10) | 65.3 | 66.4 | 66.9 | 68.0 | 77.3 | 75.2 | 64.8 | 61.3 | 68.6 | 67.7 |
| ACT ($d$=3, $k$=20) | 67.0 | 67.9 | 67.2 | 69.0 | 76.2 | 76.9 | 65.5 | 61.8 | 69.0 | 68.9 |
| ACT ($d$=4, $k$=10) | 66.3 | **71.1** | 70.5 | 70.1 | 74.3 | 70.1 | 66.7 | 66.8 | 69.5 | 69.5 |
| ACT ($d$=4, $k$=20) | **68.3** | 70.8 | 70.3 | **70.8** | 82.3 | 81.8 | 70.3 | 73.5 | 72.8 | 74.2 |
| **SPAM** | | | | | | | | | | |
| CoT (0-shot) | 73.3 | 78.6 | 95.5 | 95.3 | 95.7 | 95.5 | 93.2 | 93.6 | 89.4 | 90.8 |
| DSPy (BFSR, 8 demos) | 95.0 | 95.2 | 98.3 | 98.3 | 98.2 | 98.2 | 96.5 | 96.6 | 97.0 | 97.1 |
| TEXTGRAD | 92.6 | 93.2 | 97.3 | 97.3 | 96.7 | 96.5 | 96.0 | 96.2 | 95.7 | 95.8 |
| TF-IDF + CART ($d$=5) | —— | —— | —— | —— | —— | —— | —— | —— | 92.7 | 93.1 |
| TF-IDF + XGBoost ($d$=3, $nt$=200 ) | —— | —— | —— | —— | —— | —— | —— | —— | 96.3 | 96.3 |
| BERT | —— | —— | —— | —— | —— | —— | —— | —— | 99.5 | 99.5 |
| ROBERTA | —— | —— | —— | —— | —— | —— | —— | —— | 99.3 | 99.3 |
| Con. tag. | 83.2 | 85.5 | 64.2 | 45.8 | 82.3 | 85.0 | 62.3 | 39.6 | 73.0 | 64.0 |
| Rule Fit | | | | | | | | | 97.0 | 96.9 |
| ACT ($d$=2, $k$=5) | 95.8 | 95.8 | 98.6 | 98.5 | 98.2 | 98.2 | 96.6 | 96.8 | 97.3 | 97.3 |
| ACT ($d$=2, $k$=10) | 96.5 | 96.6 | **99.2** | **99.2** | 99.2 | 99.2 | 98.5 | **98.5** | 98.4 | 98.4 |
| ACT ($d$=3, $k$=5) | 98.2 | 98.1 | 97.0 | 96.9 | **99.4** | **99.5** | 98.5 | 98.5 | 98.3 | 98.3 |
| ACT ($d$=3, $k$=10) | **98.5** | **98.4** | 98.8 | 98.8 | 99.1 | 99.2 | **98.8** | **98.8** | 98.8 | 98.8 |
| **JAILBREAK** | | | | | | | | | | |
| CoT (0-shot) | 77.9 | 82.2 | 85.5 | 83.5 | 90.4 | 91.4 | *81.1* | 77.7 | *83.7* | 83.7 |
| DSPy (BFSR, 8 demos) | 89.6 | 89.3 | *91.6* | **91.6** | 94.8 | 94.8 | 83.5 | 83.9 | 89.9 | 89.9 |
| TEXTGRAD | **91.3** | **91.3** | 90.4 | 90.3 | 94.7 | 95.1 | *88.8* | 90.0 | *91.3* | 91.7 |
| TF-IDF + CART ($d$=4) | —— | —— | —— | —— | —— | —— | —— | —— | 92.8 | 92.4 |
| TF-IDF + XGBoost ($d$=2, $nt$=200 ) | —— | —— | —— | —— | —— | —— | —— | —— | 96.8 | 96.8 |
| BERT | —— | —— | —— | —— | —— | —— | —— | —— | 98.4 | 98.4 |
| ROBERTA | —— | —— | —— | —— | —— | —— | —— | —— | 99.6 | 99.6 |
| Con. tag. | 82.7 | 82.2 | 58.6 | 31.8 | 82.7 | 79.8 | 76.3 | 70.1 | 75.1 | 66.0 |
| Rule Fit | | | | | | | | | 96.8 | 96.8 |
| ACT ($d$=3, $k$=10) | 82.3 | 82.7 | 85.5 | 85.1 | 95.4 | 95.8 | 82.7 | 82.6 | 86.5 | 86.6 |
| ACT ($d$=3, $k$=20) | 84.3 | 86.5 | *91.2* | 90.5 | 96.2 | 96.6 | *87.6* | 86.6 | *89.8* | 89.9 |
| ACT ($d$=4, $k$=10) | 90.0 | 90.1 | *91.6* | 91.0 | 97.2 | 97.2 | 85.1 | 83.9 | *91.0* | 90.6 |
| ACT ($d$=4, $k$=20) | *90.8* | 90.0 | **92.0** | 91.5 | **98.9** | **98.8** | *92.0* | 92.7 | 93.4 | 93.3 |
| **BANKCHURN** | | | | | | | | | | |
| CoT (0-shot) | 47.8 | 48.1 | 48.5 | 25.9 | 52.8 | 60.5 | 50.2 | 37.3 | 49.8 | 43.0 |
| DSPy (BFSR, 8 demos) | 52.8 | 55.6 | 52.3 | 42.8 | 54.0 | 52.2 | 57.2 | 58.5 | 54.1 | 52.3 |
| TEXTGRAD | 52.0 | 60.0 | 53.3 | 60.3 | 55.3 | 61.1 | 52.8 | 61.4 | 53.4 | 60.6 |
| TF-IDF + CART ($d$=5) | —— | —— | —— | —— | —— | —— | —— | —— | 63.2 | 66.8 |
| TF-IDF + XGBoost ($d$=2, $nt$=200 ) | —— | —— | —— | —— | —— | —— | —— | —— | 56.7 | 48.8 |
| BERT | —— | —— | —— | —— | —— | —— | —— | —— | 71.7 | 71.6 |
| ROBERTA | —— | —— | —— | —— | —— | —— | —— | —— | 72.3 | 72.3 |
| Con. tag. | 57.2 | 60.2 | 52.0 | 59.8 | 59.7 | 60.8 | 54.3 | 61.6 | 55.8 | 60.6 |
| Rule Fit | | | | | | | | | 63.0 | 59.8 |
| ACT ($d$=3, $k$=10) | 56.3 | 42.2 | 58.8 | 67.4 | 60.5 | 64.6 | 57.5 | 62.7 | 58.3 | 59.2 |
| ACT ($d$=3, $k$=20) | 58.0 | 47.8 | **67.0** | 67.5 | **69.2** | 65.3 | **62.7** | **63.7** | 64.2 | 61.1 |
| ACT ($d$=4, $k$=10) | 61.7 | 65.3 | 61.7 | 60.9 | 69.0 | 65.0 | 60.9 | 60.2 | 63.3 | 62.9 |
| ACT ($d$=4, $k$=20) | **65.2** | **70.7** | 65.0 | **69.1** | 68.7 | **68.6** | 62.0 | 60.0 | 65.2 | 67.1 |
| **IMBD** | | | | | | | | | | |
| CoT (0-shot) | 92.3 | 91.9 | 94.2 | 94.1 | 95.7 | 95.6 | 94.2 | 94.0 | 94.1 | 93.9 |
| DSPy (BFSR, 8 demos) | 93.6 | **93.6** | **95.5** | **95.5** | **96.8** | **96.8** | 94.8 | 94.8 | 95.2 | 95.2 |
| TEXTGRAD | 93.2 | 92.9 | 95.2 | 95.1 | 95.9 | 95.8 | 94.7 | 94.5 | 94.8 | 94.6 |
| TF-IDF + CART ($d$=3) | —— | —— | —— | —— | —— | —— | —— | —— | 66.8 | 72.8 |
| TF-IDF + XGBoost ($d$=2, $nt$=200 ) | —— | —— | —— | —— | —— | —— | —— | —— | 76.2 | 76.7 |
| BERT | —— | —— | —— | —— | —— | —— | —— | —— | 90.2 | 90.1 |
| ROBERTA | —— | —— | —— | —— | —— | —— | —— | —— | 92.7 | 92.7 |
| Con. tag. | 74.8 | 76.9 | 56.2 | 31.7 | 50.0 | 66.7 | 51.3 | 63.8 | 58.1 | 59.8 |
| Rule Fit | | | | | | | | | 75.2 | 73.7 |
| ACT ($d$=3, $k$=10) | 93.2 | 93.2 | 92.8 | 92.8 | 95.7 | 95.7 | 93.8 | 93.9 | 93.9 | 93.9 |
| ACT ($d$=3, $k$=20) | **93.7** | **93.6** | 94.7 | 94.7 | 96.3 | 96.1 | **95.5** | **95.5** | 95.1 | 95.0 |

**Baselines.** We evaluate ACT against nine classification baselines:

- **Chain-of-Thought (CoT, zero-shot)** (Wei et al., 2022): standard zero-shot prompting with step-by-step reasoning, without access to labeled examples or demonstrations.
- **DSPy (8-shot in-context learning)** (Khattab et al., 2023): a demonstration-based approach that conditions the model on eight curated input–label examples to elicit task-relevant behavior.
- **TextGrad** (Yuksekgonul et al., 2024): a prompt optimization method that refines a single task-specific prompt using textual feedback. The prompt includes the task description and class labels, and is optimized for accuracy. We follow the original setup, with no demonstrations provided.
- **TF-IDF + CART**: a CART trained on TF-IDF representations of the raw texts (Salton et al., 1975; Breiman et al., 1984) that does not rely on a language model, offering structural transparency but limited semantic interpretability, as it lacks natural-language reasoning.
- **TF-IDF + XGBoost**: a gradient-boosted tree ensemble trained on TF-IDF representations of the texts (Chen & Guestrin, 2016). We tune both the maximum tree depth and the number of trees (see Appendix D, and report the best-performing configuration per dataset.
- **Fine-tuned BERT and RoBERTa classifiers**: supervised Transformer baselines initialized from pretrained BERT and RoBERTa encoders (Devlin et al., 2019; Liu et al., 2019) and fine-tuned end-to-end on each dataset for text classification. These models provide strong predictive performance references, but do not yield explicit, human-readable decision rules.
- **LLM Concept Bottleneck + CART (Conc. tag.)**: a concept-bottleneck baseline where an LLM extracts salient domain concepts and encodes each text as a binary concept vector; a shallow CART is then trained on this concept space.
- **RuleFit (TF-IDF rules)** (Friedman & Popescu, 2008): a rule-ensemble method that learns sparse, human-readable if–then rules over TF-IDF features.

The three prompt-based baseline methods (CoT, DSPy, TextGrad) are initialized with task-specific prompts that explicitly state the classification objective and define the target classes (e.g., for DIAGNO, whether a case indicates Tuberculosis). In contrast, ACT begins from a generic, task-agnostic query ("Based on the provided example, does it belong to the positive class?"), without access to the task description, class names, or any domain-specific information. Consequently, ACT must uncover the task through successive node's question-refinement iterations, making the comparison conservative with respect to considered baselines. We vary two hyperparameters of the proposed ACT algorithm: the maximum tree depth $d$ and the number of prompt refinement steps per node $k$, exploring different configurations in our experiments. To control input length and computational cost, $f_{\text{purity}}$ is restricted to at most $m$ randomly selected well-classified and $m$ randomly selected misclassified instances per group (with $m = 50$ in our experiments). To ensure a fair comparison, all methods are evaluated using identical underlying language models.

## 4.1 EXPERIMENTAL RESULTS AND DISCUSSION

**Test set accuracy** As shown in Table 2, ACT with appropriately selected hyperparameters consistently matches or surpasses the CoT, TEXTGRAD, and DSPy baselines across datasets and LLMs. Concretely, averaged over all datasets and models, the best-performing ACT configuration ($d = 4$, $k = 20$) improves test accuracy by 5.0 percentage points over DSPy (8-shot) and by 5.1 points over TEXTGRAD, while providing transparent and interpretable question-based decision paths.

Additionally, we compare our method against CART with TF-IDF features at depth 3, which exhaustively searches for optimal splits based on the TF-IDF matrix. This baseline achieves strong performance (78.8% on DIAGNO, 89.3% on SPAM, 91.3% on JAILBREAK). ACT consistently outperforms TF-IDF+CART on SPAM and JAILBREAK for both GPT-4.1 variants, while only Mini exceeds TF-IDF+CART on DIAGNO. However, the TF-IDF+CART approach remains fundamentally limited by its lack of semantic interpretability. The resulting decision trees (see, e.g., Fig. 5 and 6 in Appendix) are typically hard to understand for users, especially when compared to the natural-language questions learned with ACT (cf. Figure 1 for DIAGNO, Figures 3 and 4 for SPAM and JAILBREAK). This stems in part from TF-IDF+CART selecting single words or n-grams as decision criteria, without any regard to context or semantic meaning. As a result, these models struggle with negation, rephrasing, and adversarially crafted text—features that are particularly prevalent in

SPAM and JAILBREAK. These limitations explain ACT's consistent advantage in these domains. Furthermore, Appendix B.1 compares the most relevant symptoms for Tuberculosis identified by ACT on the DIAGNO dataset with those from established medical sources, highlighting their strong alignment and the efficacy of ACT in high-stakes scenarios.

**Interpretability** To quantify ACT's interpretability, we analyze the average path length at inference and the average node-question length. Results in Appendix F, supported by manual inspection of the resulting ACTs as shown in Appendix B.2 and Appendix B.3, provide evidence that ACT generally tends to produce interpretable classifications.

## 4.2 ABLATION STUDIES

The proposed ACT algorithm has two key hyperparameters: the depth of the tree $d$ and the number of optimization steps per node during training $k$. Additional hyperparameters with less influence on the accuracy include the number $m$ of examples $X_{correct}$ and $X_{error}$ provided to the LLM ($f_{purity}$) when performing semantic analysis (cf. Appendix E.2 for more discussion) and the maximum number of logical operators $L$ permitted in each generated question (cf. Appendix A.2 for more discussion). As illustrated in Figure 2, the depth $d$ and optimization steps $k$ significantly influence ACT performance. Across datasets and LLMs, test accuracy peaks at depths 4–5, with deeper trees exhibiting decreased test accuracy despite increased training accuracy at depth 6 —an indication of overfitting. Conversely, increasing optimization steps consistently improves performance, though gains plateau after 10–20 steps depending on the configuration, indicating that only a modest number of steps $k$ is sufficient to identify the best splitting question.

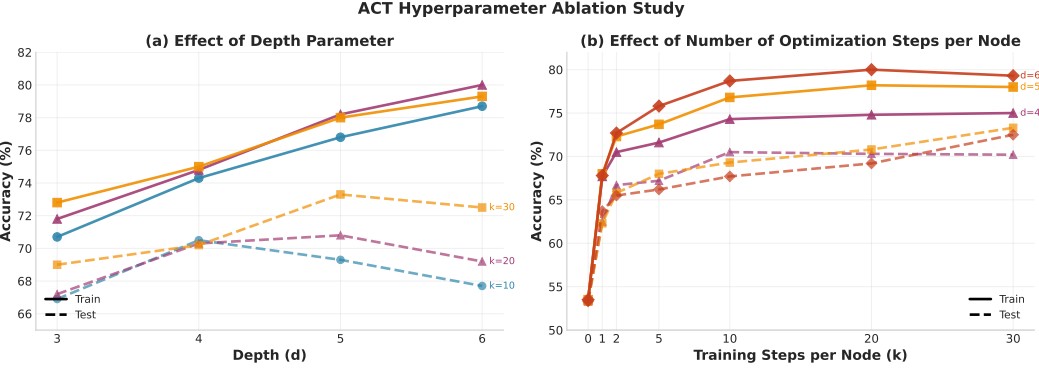

Figure 2: Ablation study of ACT hyperparameters on the DIAGNO dataset using GPT-4.1 Nano. (a) Effect of depth parameter $d$ on model accuracy for different numbers of optimization steps per node during training $k$. (b) Effect of the number of optimization steps per node during training $k$ on model accuracy for different depth values $d$. Solid lines represent training accuracy; dashed lines represent test accuracy.

## 5 CONCLUSION

We introduced the Agentic Classification Tree (ACT), a novel framework that combines the interpretability of traditional decision trees with the semantic reasoning capabilities of large language models (LLMs). Unlike conventional decision trees that rely on rigid feature-based splits, ACT dynamically optimizes natural-language prompts at each node, using LLM responses to perform semantically meaningful binary splits. Our experiments on text-based binary classification tasks demonstrate that ACT achieves competitive accuracy while providing interpretable, language-based decision logic. This approach highlights the potential of combining classical machine learning structures with modern language model reasoning, opening new avenues for interpretable and effective decision-making in complex, text-rich domains. Future work includes extending ACT to multi-class classification and regression task, improving computational efficiency through targeted prompt optimization strategies, and exploring applications beyond text classification to other structured and unstructured data modalities.

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

## A ADDITIONAL IMPLEMENTATION DETAILS

### A.1 ILLUSTRATIVE EXAMPLE: PROMPT REFINEMENT VIA SEMANTIC FEEDBACK

We describe the iterative prompt refinement procedure used in ACT, using an illustrative case from the `DIAGNO3` medical diagnosis task. At each decision node, the objective is to construct a binary natural-language question that separates the data into semantically meaningful groups, thereby reducing weighted Gini impurity.

At each iteration $k$, the following steps are performed:

1. The current prompt $p^{(k)}$ is evaluated over the training set. For each input $x_i$, the LLM outputs a binary decision (`yes`/`no`).

2. The data is partitioned into two subsets: those routed to the `yes` branch (denoted **sdL**) and those routed to the `no` branch (denoted **sdG**).

3. For each subset, the correctly classified and misclassified examples are separated. These are then passed to the LLM, which is prompted to identify semantic features that distinguish the two groups.

4. Based on this contrastive analysis, the LLM generates feedback indicating which aspects of the current prompt are ambiguous or insufficient. This feedback is then used to revise the question, yielding a new candidate $p^{(k+1)}$.

This iterative refinement continues until a stopping criterion is met (e.g., convergence or maximum number of updates). In practice, we observe that prompts evolve from generic task-agnostic formulations to more specific, domain-relevant queries. For instance, an initial prompt such as *"Does this example belong to the positive class?"* may be refined into a targeted question like *"Does the example mention coughing up blood or high fever?"*, better aligned with the semantic distinctions in the data.

**Iteration 0 — Initial Generic Prompt Prompt** $p^{(0)}$: `Based on the provided context, does this example belong to the positive class? (yes/no)`

**Gini impurity:** 0.495

**Group predicted "YES" (sdL):**

- **Correct predictions:** Examples often exhibited severe and systemic symptoms such as coughing blood, weight loss, shortness of breath, rashes, and fatigue.
- **Misclassified "no" cases:** Contained milder or localized symptoms like eye itching, mild sore throat, or isolated skin rashes, which were insufficiently discriminated by the general prompt.

**Group predicted "NO" (sdG):**

- **Correct predictions:** Mostly examples with mild and localized symptoms—itchy nose, minor pain, swelling—without strong systemic indicators.
- **Misclassified "yes" cases:** Actually showed critical signs such as coughing up blood, high fever, significant pain, or weight loss, which the current question failed to detect.

**Global semantic feedback:** *"Focusing the question on specific, highly discriminative features—such as the presence of severe systemic symptoms like coughing blood, weight loss, or high fever—can enhance class separation. This targeted approach reduces ambiguity, improves alignment with key features, and minimizes misclassification."*

**Revision** $\rightarrow$ $p^{(1)}$: `Does this example show severe systemic symptoms such as coughing blood, weight loss, or high fever? (yes/no)`

**Iteration 1 — Emphasizing a Single Discriminative Feature Prompt** $p^{(1)}$: `Does this example show severe systemic symptoms such as coughing blood, weight loss, or high fever? (yes/no)`

**Gini impurity:** 0.470

**Group predicted "YES" (sdL):**

- **Correct predictions:** Frequently involved coughing up blood alongside systemic symptoms like nasal congestion, eye itching, fatigue, fever, and skin lesions—indicating strong respiratory or infectious patterns.

- **Misclassified "no" cases:** Contained only mild respiratory issues (e.g., itchy or runny nose) and lacked the critical signal of coughing up blood. The model overgeneralized and flagged weakly indicative cases as positive.

**Group predicted "NO" (sdG):**

- **Correct predictions:** Examples were diverse but lacked specific severe indicators like high fever or respiratory distress. These cases included generalized symptoms (e.g., skin rashes, fatigue) that didn't clearly indicate the positive class.

- **Misclassified "yes" cases:** Contained classic positive indicators such as high fever, muscle pain, fatigue, and systemic lesions. The model failed to recognize the more complex symptom constellation as positive.

**Global semantic feedback:** *"The current question is too broad and includes features (like fever or weight loss) that are insufficiently discriminative. Focus the question on the most specific positive-class indicator — coughing up blood — and remove less reliable features. This should improve class separation by reducing false positives."*

**Revision** $\rightarrow p^{(2)}$: `Does this example show evidence of coughing up blood? (yes/no)`

**Iteration 2 — Broadening to Capture Severe Cases  Prompt** $p^{(2)}$: `Does this example show evidence of coughing up blood?  (yes/no)`

**Gini impurity:** 0.468

A.2    PROMPT CONSTRAINTS DURING PROMPT OPTIMIZATION

During prompt optimization in ACT, additional constraints are applied to keep questions clear and interpretable while encouraging exploration. The goal of these constraints is twofold: (i) ensure that generated questions remain simple, precise, and auditable, and (ii) incentivize diversity so that ACT explores many different semantic formulations during refinement. The maximum number of logical operators $L$ allowed per question is a controllable hyperparameter in ACT. In all experiments reported in the paper, we set $L = 2$.

---

**LLM Task: Generate Optimized Question (with Constraints)**

The following constraints must always be satisfied:

- The question has to be clear and easy.
- The question must focus on at most two **characteristics**.
- The question must include at least one **characteristic** different from the previous one.
- The content of the new question must be significantly different from the current one.
- Use at most $L$ logical operators (`and`/`or`).
- The question has to be answerable with `yes` or `no` only.
- The question must finish with "(yes/no)?".
- Do not use vague words like `could`, `might`, or `possibly`.
- Do not use blanks or placeholder tags like `\___"` or `<...>`.

---

## A.3 REVIEW OF TRADITIONAL DECISION TREE METHODS: CART AND C4.5

Classification and Regression trees (CART), proposed by Breiman et al. (1984), are widely used decision tree algorithms for supervised classification tasks. In this work, we focus exclusively on the classification setting.

In this subsection we define $\mathcal{D}$ as a tabular dataset $\mathcal{D} = \{(x_i, y_i)\}_{i=1}^{N}$, where each instance $x_i \in \mathbb{R}^d$ is associated with a target output $y_i$, CART constructs a binary decision tree by recursively partitioning the dataset based on threshold splits on numeric input features or binary partitions of categorical inputs, selected in a greedy manner. We detail these steps below.

**Node Definition.** At each internal node $x$ of the tree, we denote by $\mathcal{D}^x \subseteq \mathcal{D}$ the subset of the training data that reaches node $x$. In particular, for the root node $x_0$, we have $\mathcal{D}^{x_0} = \mathcal{D}$, the full training dataset.

To split the data at node $x$, CART evaluates candidate partitions along a feature dimension $j$ and a threshold $s \in \mathbb{R}$. The dataset $\mathcal{D}^x$ is then divided into two subsets:

$$\mathcal{D}_L^x = \{(x_i, y_i) \in \mathcal{D}^x \mid x_{i,j} \leq s\}, \quad \mathcal{D}_R^x = \{(x_i, y_i) \in \mathcal{D}^x \mid x_{i,j} > s\}. \tag{2}$$

These subsets correspond to the training data that will be used at the left and right child nodes of $x$, respectively. That is, if $y_L$ and $y_R$ denote the left and right children of node $x$, we define $\mathcal{D}^{y_L} = \mathcal{D}_L^x$ and $\mathcal{D}^{y_R} = \mathcal{D}_R^x$.

For categorical features, CART evaluates binary partitions over subsets of discrete values, typically by assigning certain categories to the left or right child nodes based on the impurity criterion.

For categorical attributes, CART partitions the data based on whether instances belong or do not belong to specific category subsets.

**Best split criterion** At each node $x$, CART selects the optimal feature and split threshold by minimizing a node-specific impurity score. This score is typically computed using the Gini impurity, which quantifies the heterogeneity of class labels within a subset.

Given a candidate split of the local dataset $\mathcal{D}^x$ into $\mathcal{D}_L^x$ and $\mathcal{D}_R^x$, the impurity of the split is defined as the weighted sum of the impurities of the two resulting subsets:

$$\delta_j^s = \frac{|\mathcal{D}_L^x(j,s)|}{|\mathcal{D}^x|} \cdot \text{Gini}(\mathcal{D}_L^x(j,s)) + \frac{|\mathcal{D}_R^x(j,s)|}{|\mathcal{D}^x|} \cdot \text{Gini}(\mathcal{D}_R^x(j,s)), \tag{3}$$

The Gini impurity of any subset $D \subseteq \mathcal{D}$ is given by:

$$\text{Gini}(D) = 1 - \sum_{k \in \mathcal{Y}} p_k^2, \tag{4}$$

where $\mathcal{Y}$ denotes the set of possible class labels, and $p_k$ is the empirical proportion of class $k$ in the dataset $D$.

A Gini impurity of 0 indicates perfect purity—i.e., all instances in the subset belong to a single class—while a value closer to 0.5 (in binary classification) indicates a highly mixed, impure subset.

**Best split search** The CART algorithm constructs the decision tree recursively using a greedy strategy: at each internal node $x$, the best split is selected by optimizing a local impurity criterion over the current subset $\mathcal{D}^x$. This split is chosen by evaluating its effect on the resulting child nodes—i.e., by computing the weighted impurity of the partition it induces. In practice, this local optimization is typically performed via exhaustive search over all candidate features and thresholds, though heuristic approximations may be used in high-dimensional settings.

The recursive construction proceeds until a stopping criterion is met, such as achieving node purity, reaching a minimum number of samples, or exceeding a predefined maximum depth $d_{\max}$.

**Tree construction.** Decision trees are constructed recursively. At each node, the split minimizing impurity is applied to partition the data, and the procedure recurses on the resulting subsets. Recursion terminates when purity is reached or a predefined stopping condition is met (e.g., maximum depth, minimum node size), and the node is labeled with the majority class.

**C4.5.** C4.5 (Salzberg, 1994) follows the same recursive tree-building process as CART but differs mainly in two respects: (i) it chooses splits by maximizing the *information gain ratio* rather than minimizing Gini impurity, and (ii) it allows multiway splits on categorical attributes, whereas CART restricts all splits to be binary.

While both CART and C4.5 have become canonical algorithms for structured data, they are not directly applicable to unstructured domains such as text or images. This limitation motivates our proposed Agentic Classification Tree (ACT).

## A.4    DETAILS ON DATASETS

We use five publicly available text classification datasets spanning diverse domains. DIAGNO, SPAM, and JAILBREAK are included because they represent tasks for which structured reasoning is expected to be beneficial, aligning with the types of problems ACT is designed to address. BANKCHURN, a tabular dataset serialized into text, is included because it relies less on pretraining-aligned task semantics and more on learning dataset-specific relationships between attributes and the churn label. IMDB, a widely used benchmark for sentiment classification, is added both to broaden the evaluation and to facilitate comparison with prior work. All datasets are relatively small and mostly balanced, reflecting the low-data prompt-optimization regime in which ACT and comparable methods are typically evaluated (Yuksekgonul et al., 2024).

- **DIAGNO** (Ninaa510, 2024): a medical diagnosis dataset (tuberculosis vs. allergic sinusitis) , a derived dataset based on DDXPlus Fansi Tchango et al. (2022). We constructed a balanced subset with 600 training, 100 validation, and 600 test samples, ensuring a 50/50 split (300 tuberculosis and 300 allergic sinusitis cases).
- **SPAM**: an email spam detection dataset derived from Deysi (2023). We constructed a balanced subset with 600 training, 100 validation, and 600 test samples, following the same procedure as for DIAGNO.
- **JAILBREAK** (Shen et al., 2024): a jailbreak prompt classification dataset with 923 training, 102 validation, and 249 test samples (32 examples have been dropped due to high context tokens).
- **BANKCHURN** (Topre, 2025): a customer–attrition prediction dataset originally provided in tabular form. Following Arzaghi et al. (2025), we serialize each record into a short textual profile describing the customer attributes. We constructed a balanced subset with 600 training, 100 validation, and 600 test samples, following the same procedure as for DIAGNO.
- **IMDB** (NLP, 2020): a sentiment analysis dataset consisting of movie reviews labeled as positive or negative. We constructed a balanced subset with 600 training, 100 validation, and 600 test samples, following the same procedure as for DIAGNO.

# B    ADDITIONAL QUALITATIVE RESULTS

## B.1    QUALITATIVE ANALYSIS OF THE QUESTIONS GENERATED BY ACT

Table 3: Symptoms comparison between ACT and medical sources.

| Questions in common | ACT only | Medical sources only |
| --- | --- | --- |
| Coughing up blood | Shortness of breath | Chest pain |
| Fever | Severe pain | Headache |
| Weight loss | | Increased sweating |
| Swollen lymph nodes | | Loss of appetite |
| Fatigue | | |

In addition to its tree structure, another crucial element ensuring the interpretability of ACT lies in how relevant the generated questions are. For this purpose, we propose to compare these questions with the ones that domain experts would have asked, focusing on the DIAGNO dataset. We survey several relevant medical sources to collect a list of symptoms that are commonly used for tubercu-

losis diagnosis. These sources include websites of the World Health Organization[5] and the British National Health Service[6], and a literature review on tuberculosis diagnosis Storla et al. (2008).

Table 3 shows a side-by-side comparison of the symptoms identified by ACT as positively associated with tuberculosis, and those most commonly identified in the aforementioned medical sources, differentiating the symptoms identified by both ACT and the medical sources (left column), from those that were exclusive (center and right columns). We see that most of the symptoms appear in common. Some of the differences may be explained by ambiguity between some symptoms (e.g. "severe pain" vs. Chest pain and headaches). Others may come from binary classification setting considered, opposing tuberculosis cases to allergy, potentially resulting in some symptoms (e.g. loss of appetite) not being represented in the dataset.

Overall, this tends to show that ACT is indeed able to identify relevant symptoms to make its predictions, showcasing its utility and reliability.

## B.2 AGENTIC CLASSIFICATION TREE FOR SPAM DATASET

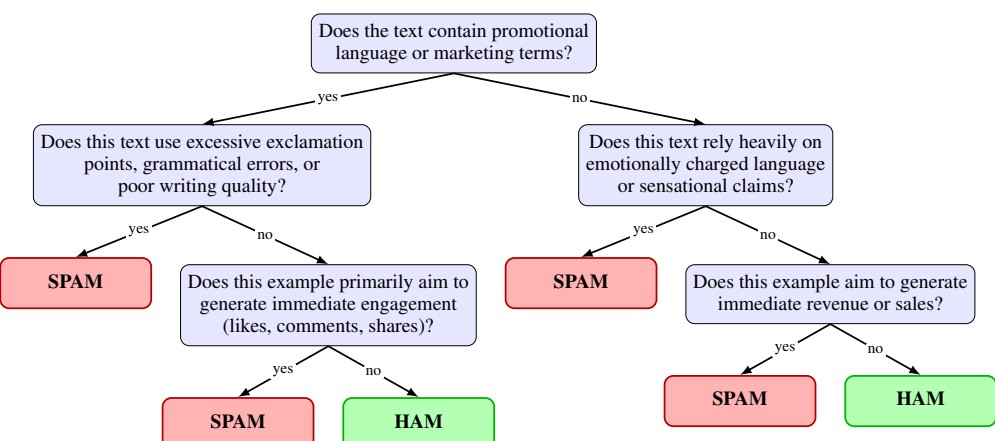

Figure 3: Decision tree of depth 3 generated by the Gemma3-4B model for spam email classification. The tree distinguishes between spam and legitimate email (ham) through hierarchical semantic questions about promotional content, writing quality, and intent. Each internal node represents a binary question optimized through the ACT framework to maximize class separation, with leaf nodes indicating the final classification based on the majority class of training examples.

---

[5]https://www.who.int/news-room/fact-sheets/detail/tuberculosis
[6]https://www.nhs.uk/conditions/tuberculosis-tb/

### B.3 Agentic Classification Tree for Jailbreak Dataset

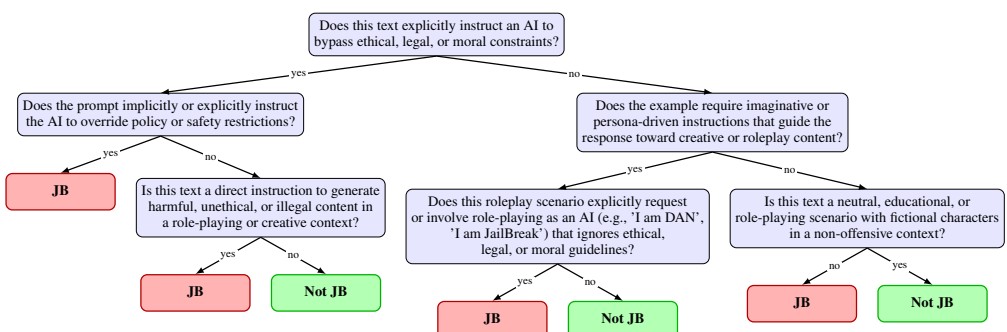

Figure 4: Decision tree of depth 3 generated by the Qwen3-4b model for jailbreak prompt classification. The tree recursively partitions inputs through binary natural-language questions optimized to distinguish between jailbreak attempts (JB) and legitimate prompts (Not JB). Each internal node contains a semantically meaningful question discovered through iterative prompt refinement, with terminal nodes indicating the final classification based on the majority class of training examples reaching that leaf.

### B.4 Qualitative comparison with TF-IDF + CART

In this section we aim to illustrate the benefits of leveraging LLM agents to directly deal with unstructured data by comparing ACT with the traditional TF-IDF and CART combination. The tree resulting from this procedure for the DIAGNO dataset is shown in Figure 5, and in Figure 6 for the SPAM dataset. In both cases, we observe the limitation of this approach, as nodes built only on one word provide little help in understanding the model. Worse, these words may often end up being generic adverbs and prepositions (e.g. "our", "any"), resulting ultimately in explanations that are both hard to understand and leverage.

On the other hand, the decision trees learned by ACT for these datasets can be easily understood, and verified, by a user.

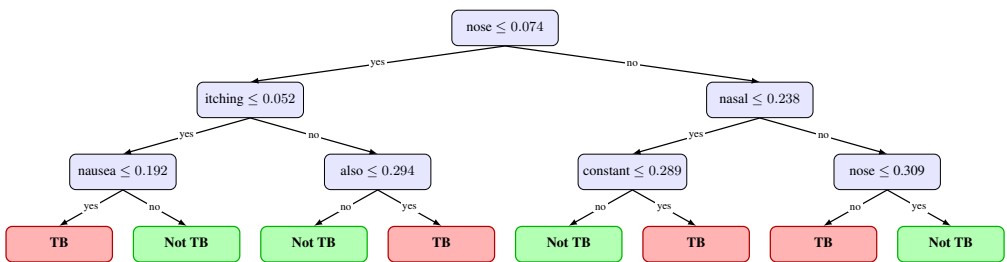

Figure 5: Decision tree of depth 3 generated by training a CART model after performing a TF-IDF preprocessing step on the DIAGNO dataset.

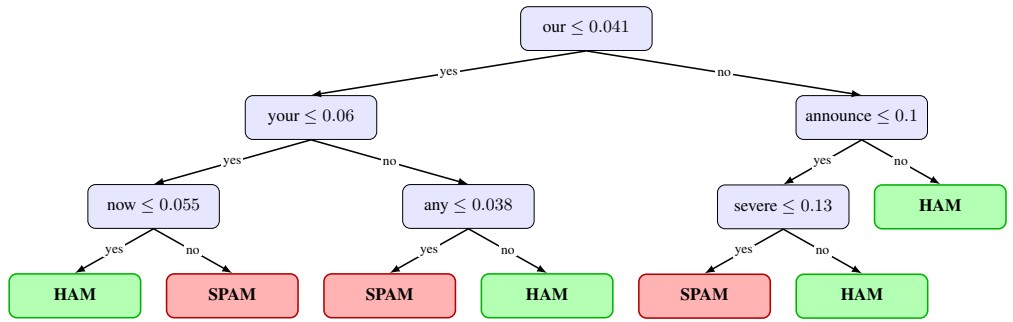

Figure 6: Decision tree of depth 3 generated by training a CART model after performing a TF-IDF preprocessing step on the SPAM dataset.

## C  COMPUTATIONAL COST ANALYSIS

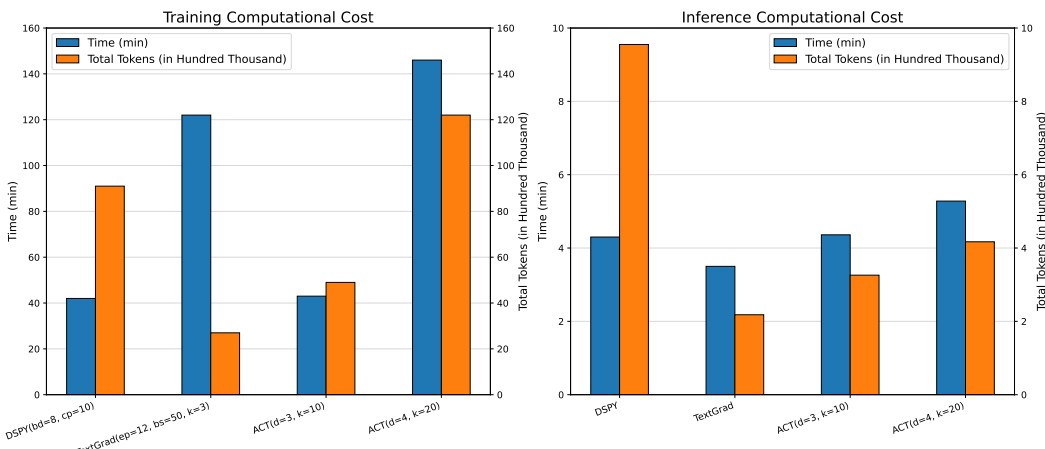

Figure 7: **Training and inference computational cost comparison on DIAGNO.** Each plot reports the average wall-clock time and token usage over 5 runs for TextGrad, DSPy (BFSR, 8-shot), and ACT under both the cheapest ($d = 3$, $k = 10$) and most expensive ($d = 4$, $k = 20$) configurations, using Gemma-3-4B on a single H200 GPU with 8 threads per program. Training cost reflects program construction dynamics: TextGrad processes fewer but very long prompts; DSPy evaluates 13 medium-sized candidate programs derived from few-shot traces; and ACT issues many short node-level prompts for Gini estimation supplemented with a smaller number of larger prompts for semantic feedback and node optimization. Inference cost shows that TextGrad is the most economical, while ACT remains within the same order of magnitude and provides higher accuracy with more interpretable outputs. DSPy exhibits similar wall-clock time to ACT but requires more tokens, largely due to the substantial size of the few-shot prompts.

## D  ADDITIONAL CLASSICAL BASELINES

To complement the prompting-based baselines used in the main paper, we report in Table 4 an extended comparison with purely symbolic lexical models. These include TF–IDF + CART with depths ranging from 2 to 5, and TF–IDF + XGBoost with multiple depth and tree-count configurations. These models provide a transparent, non-LLM control group and clarify the performance gap between ACT and traditional supervised baselines across all five datasets. As shown below, ACT remains competitive or superior to these methods while offering semantically grounded, human-readable decision rules.

Table 4: Performance of TF-IDF + CART and TF-IDF + XGBoost across all datasets.

| Dataset / Split | CART d=2 | CART d=3 | CART d=4 | CART d=5 | XGB (2,50) | XGB (2,200) | XGB (3,50) | XGB (3,200) | XGB (4,50) | XGB (4,200) |
|---|---|---|---|---|---|---|---|---|---|---|
| DIAGNO – Train | 82.0 | 84.3 | 86.5 | **88.8** | 86.2 | 94.7 | 89.2 | 98.7 | 93.5 | **99.8** |
| DIAGNO – Test | 77.5 | **78.8** | 73.7 | 76.3 | 81.0 | 81.5 | 81.8 | 81.7 | 82.2 | **83.0** |
| SPAM – Train | 87.8 | 92.3 | 93.5 | **94.7** | 97.8 | 99.7 | 98.7 | 99.8 | 99.2 | **100.0** |
| SPAM – Test | 85.8 | 89.3 | 90.7 | **92.7** | 94.3 | 96.2 | 93.8 | **96.3** | 94.5 | 96.0 |
| JAILBREAK – Train | 88.3 | 91.1 | 93.1 | **94.3** | 95.6 | 97.9 | 96.5 | 98.5 | 97.1 | **99.0** |
| JAILBREAK – Test | 86.4 | 92.4 | **92.8** | 91.9 | 95.2 | **96.8** | 95.2 | 96.4 | 95.2 | **96.8** |
| BANKCHURN – Train | 59.7 | 63.0 | 63.7 | **66.7** | 71.5 | 84.5 | 79.3 | 90.0 | 84.2 | **93.2** |
| BANKCHURN – Test | 62.7 | 62.8 | 62.8 | **63.2** | 53.2 | **56.7** | 54.3 | 53.7 | 54.8 | 55.3 |
| IMDB – Train | 68.2 | 71.0 | 71.8 | **76.2** | 83.5 | 97.3 | 89.0 | 99.3 | 94.7 | **99.8** |
| IMDB – Test | 66.8 | **66.8** | 66.5 | **66.8** | 72.5 | **76.2** | 73.3 | 74.8 | 74.8 | 75.5 |

# E  ADDITIONAL ABLATION STUDIES

## E.1  MODEL SCALING ANALYSIS

To assess how model strength affects different prompting strategies, we conduct an additional scaling study using GPT-4.1. We evaluate CoT, DSPy, TextGrad, and two ACT configurations on the DIAGNO and BANKCHURN datasets, as these are the most challenging tasks. As shown in Table 5, stronger models do not consistently improve CoT, DSPy, or TextGrad, whereas ACT benefits from increased model capacity. These results indicate that ACT more effectively leverages the enhanced reasoning abilities of frontier models.

| Method | GPT-4.1-nano | GPT-4.1-mini | GPT-4.1 |
|---|---|---|---|
| **DIAGNO** | | | |
| CoT | 63.3 | 61.2 | 62.2 |
| DSPy | 68.7 | 64.5 | 68.2 |
| TextGrad | 64.5 | 65.8 | 64.8 |
| ACT (3, 10) | 66.9 | **77.3** | 80.2 |
| ACT (3, 20) | **67.2** | 76.2 | **81.8** |
| **BANKCHURN** | | | |
| CoT | 48.5 | 52.8 | 57.2 |
| DSPy | 52.3 | 54.0 | 59.5 |
| TextGrad | 53.3 | 55.3 | 58.2 |
| ACT (3, 10) | 58.8 | 60.5 | 69.2 |
| ACT (3, 20) | **67.0** | **69.2** | **72.5** |

Table 5: Performance scaling of CoT, DSPy, TextGrad, and ACT when moving from GPT-4.1-nano to GPT-4.1-mini to GPT-4.1. ACT benefits more consistently from stronger models.

## E.2  ABLATION ON THE NUMBER OF FEEDBACK SAMPLES $m$

We conduct an ablation study on the number of labeled samples $m$ used to generate semantic feedback during node-level prompt optimization. We evaluate $m \in \{5, 10, 20, 50\}$ on the DIAGNO dataset using GPT-4.1-nano for two ACT configurations: $(d = 3, k = 10)$ and $(d = 4, k = 20)$. The results in Tables 6 and 7 show how varying $m$ affects training performance, test metrics, and token usage. As comparison, the results for DSPy are also shown.

| $m$ | Train Acc ↑ | Test Acc ↑ | Test F1 ↑ | Tokens ($\times 100k$) ↓ |
|---|---|---|---|---|
| 5 | 69.8 | 67.0 | 67.6 | **52.4** |
| 10 | 71.2 | 67.5 | 71.3 | 53.5 |
| 20 | 71.3 | **69.5** | **71.7** | 54.2 |
| 50 | 71.8 | 67.2 | 69.0 | 55.2 |
| DSPy | 71.5 | 68.7 | 61.2 | 122.1 |

Table 6: Effect of varying $m$ for ACT $d = 3, k = 10$ on DIAGNO with GPT-4.1-nano.

| m | Train Acc ↑ | Test Acc ↑ | Test F1 ↑ | Tokens ($\times 100k$) ↓ |
|---|---|---|---|---|
| 5 | 71.7 | 65.8 | 65.9 | 124.7 |
| 10 | 74.2 | 68.7 | 70.6 | 135.0 |
| 20 | 74.5 | 70.0 | **71.3** | 137.6 |
| 50 | 74.8 | **70.3** | 70.8 | 143.8 |
| DSPy | 71.5 | 68.7 | 61.2 | **122.1** |

Table 7: Effect of varying $m$ for ACT ($d = 4, k = 20$) on DIAGNO with GPT-4.1-nano.

## F  INTERPRETABILITY STATISTICS FOR ACT

In Table 8 we report two interpretability indicators for ACT across all datasets and ACT configurations evaluated: the average question length (in characters) and the average path length (number of nodes traversed during inference). These values are averaged over the four LLMs used in our experiments.

Across datasets, average question lengths range from 78 to 155 characters, reflecting concise node-level prompts, while average path lengths show that predictions typically require only about three nodes, even for depth-4 trees. We also observe a mild trade-off in which datasets with longer questions (e.g., JAILBREAK and SPAM) yield shorter paths, indicating that ACT can consolidate multiple relevant features into a single informative query. For reference, the generic seed question *"Based on the provided example, does it belong to the positive class? (yes/no)"* has a length of 87 characters.

| ACT (d, k) | Avg. Q. Len | Avg. Path |
|---|---|---|
| (3, 10) | 92.00 | 2.35 |
| (3, 20) | 93.15 | 2.51 |
| (4, 10) | 78.05 | 3.03 |
| (4, 20) | 80.53 | 3.06 |

(a) DIAGNO

| ACT (d, k) | Avg. Q. Len | Avg. Path |
|---|---|---|
| (2, 5) | 118.00 | 1.33 |
| (2, 10) | 142.88 | 1.26 |
| (3, 5) | 94.18 | 1.36 |
| (3, 10) | 122.63 | 1.14 |

(b) SPAM

| ACT (d, k) | Avg. Q. Len | Avg. Path |
|---|---|---|
| (3, 10) | 136.00 | 1.98 |
| (3, 20) | 155.00 | 2.13 |
| (4, 10) | 142.00 | 2.41 |
| (4, 20) | 150.33 | 2.57 |

(c) JAILBREAK

| ACT (d, k) | Avg. Q. Len | Avg. Path |
|---|---|---|
| (3, 10) | 90.50 | 2.92 |
| (3, 20) | 89.90 | 2.93 |
| (4, 10) | 82.20 | 3.10 |
| (4, 20) | 89.77 | 2.96 |

(d) BANKCHURN

| ACT (d, k) | Avg. Q. Len | Avg. Path |
|---|---|---|
| (3, 10) | 81.30 | 1.97 |
| (3, 20) | 95.80 | 2.28 |

(e) IMDB

Table 8: Interpretability statistics across datasets.

## G  COST-NORMALIZED ANALYSIS

Figure 8 visualizes the accuracy–cost trade-off across methods and model sizes, highlighting how performance scales with increasing computational expense. This section further reports token usage, monetary cost, test accuracy, and cost-normalized accuracy for each method, with Table 9 summarizing these metrics across the GPT-4.1 family on DIAGNO to facilitate comparison of efficiency and scalability.

Table 9 shows that TextGrad achieves the highest cost-normalized accuracy in the Nano and Mini settings, while ACT (3,10) remains competitive—ranking just behind TextGrad in smaller models

and performing strongly on GPT-4.1. The (3,20) variant is less cost-efficient due to higher token usage, though it still slightly outperforms DSPy on Mini and GPT-4.1. As shown in Figure 8, ACT's test accuracy consistently scales with cost across all model sizes, whereas TextGrad and DSPy do not exhibit consistent cost-dependent accuracy gains.

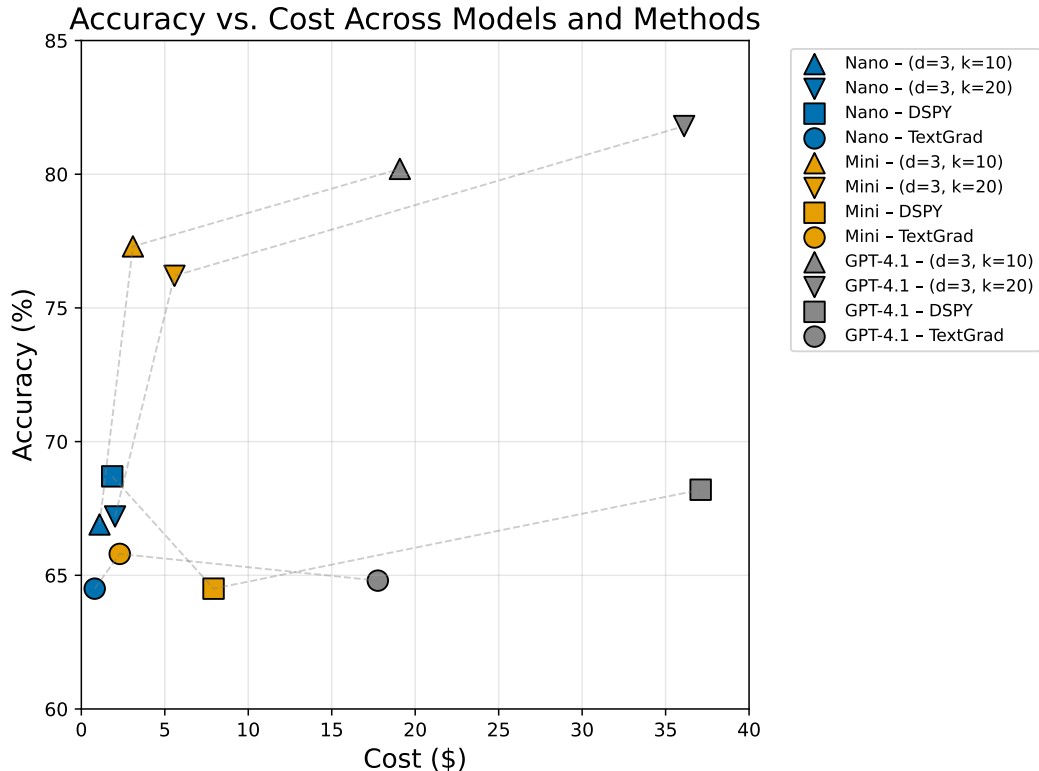

Figure 8: **Accuracy–cost trade-off across methods and model scales.** The plot shows test accuracy versus monetary cost for ACT, TextGrad, and DSPy across the Nano, Mini, and GPT-4.1 model families. Each method is connected across model scales to illustrate how performance changes with increasing cost.

Table 9: Cost normalization and model scaling across the GPT-4.1 family on the DIAGNO dataset.

| Methods | Model | Cost ↓ | Test Acc. ↑ | Total Tokens (x100) ↓ | Cost Norm Acc. ↑ |
|---|---|---|---|---|---|
| 3, 10 | Nano | 1.09 | 66.90 | 55.18 | 61.54 |
| 3, 20 | Nano | 2.02 | 67.20 | 101.37 | 33.31 |
| DSPY | Nano | 1.85 | 68.70 | 152.58 | 37.23 |
| TextGrad | Nano | 0.80 | 64.50 | 48.06 | 80.93 |
| 3, 10 | Mini | 3.09 | 77.30 | 48.08 | 24.98 |
| 3, 20 | Mini | 5.58 | 76.20 | 87.58 | 13.65 |
| DSPY | Mini | 7.92 | 64.50 | 161.28 | 8.15 |
| TextGrad | Mini | 2.30 | 65.80 | 36.37 | 28.60 |
| 3, 10 | GPT-4.1 | 19.08 | 80.20 | 53.71 | 4.20 |
| 3, 20 | GPT-4.1 | 36.11 | 81.80 | 102.46 | 2.26 |
| DSPY | GPT-4.1 | 37.10 | 68.20 | 153.89 | 1.84 |
| TextGrad | GPT-4.1 | 17.76 | 64.80 | 49.08 | 3.65 |

## APPENDIX: USE OF LANGUAGE MODELS

During the preparation of this manuscript, we made limited use of large language models to enhance the clarity and readability of the text. This involved assistance with grammar, phrasing, and stylistic improvements, particularly in the abstract and selected explanatory sections. All scientific content, including the formulation of research questions, experimental design, results, and interpretations, was developed solely by the authors. No language model was used to generate original ideas, proofs, or analyses.

