# OpenReview forum: "ACT: AGENTIC CLASSIFICATION TREE"
_ICLR.cc/2026/Conference — Submitted to ICLR 2026_

### Official Review · Reviewer_2SDA · 2025-10-30

**Soundness:** 3
**Presentation:** 3
**Contribution:** 3
**Rating:** 8
**Confidence:** 2

**Summary:**

This paper proposes Agentic Classification Trees: interpretable classifiers whose nodes are natural-language binary questions answered by an LLM. Splits combine Gini impurity with LLM semantics feedback via prompt optimization. On three binary text tasks, the method matches or exceeds prompting baselines while yielding readable decision paths.

**Strengths:**

Introduces tree-structured classification where nodes are optimized natural-language questions. Combines classical impurity metrics with LLM semantic feedback in a single framework.
Offers a path toward auditable, interpretable-by-design classifiers for unstructured data.

**Weaknesses:**

Computational cost is not analyzed.

**Questions:**

Training involves nested iterations with multiple model calls per node. Can you report call counts and token usage? How will the performance change with stronger frontier models?

---

> ### Author Response · Authors · 2025-11-22
>
> We thank the reviewer for the positive feedback on our paper, and their insightful comments that helped us improve the paper.
>
> >  Computational cost is not analyzed. (Q1) Training involves nested iterations with multiple model calls per node. Can you report call counts and token usage?
>
> We thank the reviewer for highlighting this important aspect and fully agree that a detailed analysis of computational cost is necessary. We have included an extended version of this analysis in Appendix C of the revised paper. To address the concern here, the **Computational Cost Table in the general response** reports the average wall-clock time and token usage on DIAGNO using Gemma-3-4B for both training and inference, together with additional context to help interpret these results. Overall, ACT shows comparable wall-clock time and token count to TextGrad and DSPy at both training and inference, while providing interpretability and higher accuracy.
>
> >  How will the performance change with stronger frontier models?
>
> To address this question, we conducted additional experiments with GPT-4.1 to analyze how performance scales with stronger models from the same family. We evaluated GPT-4.1 on the three main baselines (CoT, DSPy, TextGrad) and on two ACT configurations (depth-3 with 10 and 20 optimization steps per node) for the DIAGNO and BANKCHURN datasets. We focused on these datasets because they are the most challenging; for the others, the strongest smaller model (GPT-4.1-mini) already saturated.
> For DIAGNO, consistent with the comparison between GPT-4.1-mini and GPT-4.1-nano, GPT-4.1 surprisingly does not improve performance for CoT, DSPy, or TextGrad, but does improve performance for ACT. For BANKCHURN, we observe similar gains across ACT and the baselines when moving from GPT-4.1-nano → GPT-4.1-mini → GPT-4.1. Overall, these results suggest that ACT is better able to leverage the enhanced reasoning capabilities of stronger models compared to other prompt-engineering techniques.
> We included the tables below in Appendix E.1 of the revised paper, as it provides a valuable analysis of model scaling. The values in the tables are test-set accuracies.
>
>
> | Method        | GPT-4.1-nano | GPT-4.1-mini | GPT-4.1 |
> |---------------|---------------|--------------|---------|
> | **DIAGNO**    |               |              |         |
> | CoT           | 63.3          | 61.2         | 62.2    |
> | DSPy          | **68.7**          | 64.5         | 68.2    |
> | TextGrad      | 64.5          | 65.8         | 64.8    |
> | ACT (3, 10)   | 66.9          | **77.3**     | 80.2    |
> | ACT (3, 20)   | 67.2      | 76.2         | **81.8** |
> | **BANKCHURN** |               |              |         |
> | CoT           | 48.5          | 52.8         | 57.2    |
> | DSPy          | 52.3          | 54.0         | 59.5    |
> | TextGrad      | 53.3          | 55.3         | 58.2    |
> | ACT (3, 10)   | 58.8          | 60.5         | 69.2    |
> | ACT (3, 20)   | **67.0**      | **69.2**     | **72.5** |
>
> Finally, the newly added Figure 8 in Appendix G shows that ACT’s test accuracy scales consistently with cost across model sizes, whereas TextGrad and DSPy do not exhibit comparable cost-dependent gains.

---

### Official Review · Reviewer_W3X5 · 2025-10-31

**Soundness:** 3
**Presentation:** 3
**Contribution:** 3
**Rating:** 4
**Confidence:** 4

**Summary:**

The paper introduces ACT, a decision-tree-like classification framework designed for unstructured text inputs. In this method, each split in the tree is expressed as a binary natural-language question that is answered by an LLM. These questions are iteratively improved using TextGrad to reduce impurity, and the final set of optimized questions is organized into a full decision tree for inference (see Algorithm 1). Experiments on three binary text-classification tasks: medical diagnosis, spam detection, and jailbreak detection, in which authors uses four small LLMs show that ACT often performs as well as or better than several baselines, including CoT prompting, DSPy (8-shot), TextGrad, and TF-IDF + CART. Ablation studies further demonstrate how tree depth and the number of refinement steps balance accuracy and overfitting (Table 2, Fig. 2).

**Strengths:**

1. The paper sets a clear and well-motivated goal: to develop transparent and auditable decision procedures for unstructured text, where each decision path can be directly inspected and interpreted by humans.

2. The method presents a technically elegant integration of decision tree learning and prompt optimization, in which each node in the tree is discovered through impurity-based selection combined with semantic feedback from the LLM, and once optimized, these nodes are formulated as fixed binary natural-language questions that define the decision structure.

3. The qualitative decision trees produced for the SPAM and JAILBREAK datasets are clear and easily interpretable, providing tangible evidence for the paper’s claim of model transparency and interpretability. As shown in Appendices B.3–B.4, the resulting trees present concise, human-readable question-answer paths that make the classification rationale explicit and auditable.

**Weaknesses:**

1. The datasets used in the study are small and primarily balanced, functioning more as toy text-classification tasks rather than realistic, high-stakes applications. For example, the DIAGNO and SPAM datasets contain only about 600 training samples each, which limits the generalizability of the results. Moreover, the experimental comparisons are incomplete: they exclude strong fine-tuned text classifiers such as BERT and RoBERTa, as well as interpretable rule-based systems and modern concept-bottleneck models that are directly relevant baselines for the paper’s stated goal of interpretable text reasoning (see Appendix B.1).

2. The paper does not provide any cost, latency, or resource accounting for the proposed method. Each node in the decision tree involves multiple LLM queries for impurity evaluation and semantic feedback, along with iterative TextGrad refinement steps, all of which are likely to require substantial token and compute budgets. However, no quantitative measurements of either training or inference cost and latency are reported anywhere in the paper (see section 3 & 4), making it difficult to assess the method’s practical efficiency and scalability.

3. There is a data exposure imbalance between ACT and its baselines. While DSPy is restricted to only 8 demonstration examples, ACT exposes the LLM to many more labeled samples during both node construction and refinement. Specifically, the impurity function $f_{purity}$ uses up to m=50 correct and 50 incorrect examples per group for feedback, meaning each refinement round involves substantial additional supervision. This discrepancy makes the comparison not strictly apples-to-apples, and the reported accuracy gains in Table 2 and section 4 (“m = 50”) may be partly attributable to the greater effective data access rather than to the algorithmic innovation itself.

**Questions:**

I suggest that the authors might report F1-score, ROC-AUC, and calibration measures, as well as interpretability indicators such as average path length, node coherence, and results from a user study on decision contestability. Moreover, The paper should report and control data and compute exposure across methods by detailing the number of tokens, examples, and LLM calls for ACT, DSPy, and TextGrad. It should also include ablations with smaller $m$ and fewer refinement steps to match the 8-shot baseline. Finally, report cost-normalized results, which can include tokens, runtime, and dollar cost per training run and per inference example—to clarify ACT’s efficiency and scalability.

---

> ### Author Response · Authors · 2025-11-22
>
> We thank the reviewer for their  insightful comments which helped us improve the paper. We answer each specific concern and question below.
>
> > The datasets used in the study are small and primarily balanced, functioning more as toy text-classification tasks rather than realistic, high-stakes applications. For example, the DIAGNO and SPAM datasets contain only about 600 training samples each, which limits the generalizability of the results. Moreover, the experimental comparisons are incomplete: they exclude strong fine-tuned text classifiers such as BERT and RoBERTa, as well as interpretable rule-based systems and modern concept-bottleneck models that are directly relevant baselines for the paper’s stated goal of interpretable text reasoning (see Appendix B.1).
>
> **Dataset size and scope.**
> We thank the reviewer for this observation. The datasets we use are intentionally small because ACT, like other prompt-optimization methods, operates in a regime where computational cost grows quickly with the number of labeled examples. This follows prior work: for example, the original TextGrad paper uses 12 iterations with a batch size of 3—so the model effectively sees only about 36 training examples (sampled with replacement) before validation.
>
> In our experiments, we use a more generous configuration (batch size 50 and 12 epochs, each consisting of 3 optimization steps), allowing the TextGrad baseline to optimize the prompt 36 times. This increases the number of optimization steps compared to the original setup while still operating within a low-data prompt-optimization regime.
>
> To clarify these points, we expanded the discussion of all five datasets and our rationale for their inclusion in Appendix A.4 of the revised paper.
>
> **Additional baselines.**
> - Regarding concept-bottleneck baselines, we implemented an LLM-based concept pipeline following the reviewer’s suggestion. Concretely, we (i) use an LLM to extract a set of salient domain concepts from the training texts, (ii) map each sample to a binary concept-presence vector using the LLM, and (iii) train a shallow CART classifier on these concepts space (see new baseline Conc. tag. in Table 2). Across our tasks, this LLM concept-bottleneck model performs worse than ACT.
>
>
> - As suggested by the reviewer, we additionally include a rule-based baseline.
> We use RuleFit~[1], a standard rule-based classifier that learns sparse, human-readable if–then rules over TF-IDF features. The results show that RuleFit performs competitively on some datasets but generally remains below the best fine-tuned encoders and below ACT on average.
>
>
> [1] Friedman, Jerome H., and Bogdan E. Popescu. "Predictive learning via rule ensembles." (2008): 916-954.
>
>
> > The paper does not provide any cost, latency, or resource accounting for the proposed method. Each node in the decision tree involves multiple LLM queries for impurity evaluation and semantic feedback, along with iterative TextGrad refinement steps, all of which are likely to require substantial token and compute budgets. However, no quantitative measurements of either training or inference cost and latency are reported anywhere in the paper (see section 3 and 4), making it difficult to assess the method’s practical efficiency and scalability.
>
> **Cost and resource accounting.**
> We thank the reviewer for highlighting this important aspect and fully agree that a detailed cost analysis is necessary. We now include Figure 7 in Appendix C, reporting wall-clock time and token usage for training and inference. The **Computational Cost Table in the general response** also provides average wall-clock time and token counts for both training and inference on DIAGNO using Gemma-3-4B, together with additional context to help interpret these results. Overall, ACT exhibits computational costs comparable to TextGrad and DSPy while offering higher accuracy and interpretability.

---

> ### Author Response · Authors · 2025-11-22
>
> > There is a data exposure imbalance between ACT and its baselines. While DSPy is restricted to only 8 demonstration examples, ACT exposes the LLM to many more labeled samples during both node construction and refinement. Specifically, the impurity function uses up to m=50 correct and 50 incorrect examples per group for feedback, meaning each refinement round involves substantial additional supervision. This discrepancy makes the comparison not strictly apples-to-apples, and the reported accuracy gains in Table 2 and section 4 (“m = 50”) may be partly attributable to the greater effective data access rather than to the algorithmic innovation itself.  (Q4) It should also include ablations with smaller and fewer refinement steps to match the 8-shot baseline.
>
> **Data Usage**
> Clarifying data usage is indeed important. As shown in the Computational Cost Table of the general response, the total number of tokens used by DSPy and ACT is comparable. Both methods rely on the full training set during training; however, they leverage the data differently.
> * DSPy generates a reasoning trace and answer for every training example, filters for correctly solved instances, constructs 13 candidate few-shot prompts (each with up to 8 demonstrations), and evaluates each candidate on the full training set—following the configuration used in the TextGrad paper [1].
> * ACT, in contrast, optimizes the question at each node using feedback from up to 2m labeled samples and evaluates candidate questions on all training samples reaching that node, selecting the one with the lowest Gini impurity.
>
> ACT matches or surpasses DSPy in accuracy and F1 across all five evaluated datasets, while also providing more interpretable decision paths and maintaining similar overall computational cost. Importantly, at inference time, all ACT configurations use substantially fewer tokens than DSPy (see the Computational Cost Table in the general response or Figure 7 in Appendix C), reinforcing that the two methods have comparable computational complexity and form a fair basis for comparison — especially in light of the m-ablation presented below.
>
> We also thank the reviewer for suggesting an ablation on the parameter m. We performed this study with $m \in \{5, 10, 20, 50\}$ on GPT-4.1-nano for the DIAGNO dataset using ACT configurations (d=3, k=10) and (d=4, k=20). During training, this corresponds to using at most 10, 20, 40, and 100 labeled samples when generating semantic feedback.
>
> The results (shown below and included in Appendix E.2 in the revised paper) indicate that for ACT (d=3, k=10), using fewer samples can even improve test accuracy, suggesting mild overfitting when too many examples are provided. Larger values of m do not noticeably increase the token count for this configuration, as only the input prompt in the semantic-feedback step becomes longer. For ACT (d=4, k=20), increasing m consistently improves accuracy but at higher computational cost.
>
> Overall, these ablations show that ACT is comparable in computational cost to DSPy while achieving higher accuracy and interpretability.
>
> ## Table 6 - Ablation of m for ACT (d=3, k=10) on DIAGNO with GPT-4.1-nano
> | m  | Train Acc | Test Acc | Test F1 | Total Tokens (×100k) |
> |----|-----------|----------|---------|----------------|
> | 5  | 69.8      | 67.0     | 67.6    |**52.4**        |
> | 10 | 71.2      | 67.5     | 71.3    | 53.5           |
> | 20 | 71.3      |**69.5**  |**71.7** | 54.2           |
> | 50 | 71.8      | 67.2     | 69.0    | 55.2           |
> | DSPy |  71.5   | 68.7     | 61.2    | 122.1          |
>
> ## Table 7 - Ablation of m for ACT (d=4, k=20) on DIAGNO with GPT-4.1-nano
> | m  | Train Acc | Test Acc | Test F1 | Total Tokens (×100k) |
> |----|-----------|----------|---------|----------------|
> | 5  | 71.7      | 65.8     | 65.9    | 124.7          |
> | 10 | 74.2      | 68.7     | 70.6    | 135.0          |
> | 20 | 74.5      | 70.0     |**71.3** | 137.6          |
> | 50 | 74.8      |**70.3**  | 70.8    | 143.8          |
> | DSPy |  71.5   | 68.7     | 61.2    |**122.1**       |
>
>
>
> [1] Yuksekgonul, M., Bianchi, F., Boen, J., Liu, S., Huang, Z., Guestrin, C., and Zou, J. (2024). TextGrad: Automatic “Differentiation” via Text. arXiv. <https://doi.org/10.48550/arXiv.2406.07496>

---

> ### Author Response · Authors · 2025-11-22
>
> > (Q1) Authors might report F1-score, ROC-AUC, and calibration measures
>
> **Additional metrics.**
> We agree that additional metrics can provide valuable insight, and we therefore re-evaluated all baselines and ACT configurations to report F1-scores. The **F1-scores can be found in the updated Table 2**, with some additional comments in the general response explaining the changes done in Table 2. Regarding the F1-Score, ACT beats the 4 LLM dependent baselines (CoT, TextGrad, DSPy, Concept tagging) in 15 out of 20 LLM-dataset combinations, showing that not only in test accuracy but also in F1-Score ACT performs strongly.
>
> While ACT can produce leaf-level class probabilities, baselines like CoT, DSPy, and TextGrad output only discrete labels, making ROC-AUC and calibration not directly comparable. We thus report accuracy and F1-score, which can be computed consistently across all methods.
>
> > (Q2) interpretability indicators such as average path length, node coherence, and results from...
>
> **Interpretability indicators.**
> We agree that additional quantitative evidence supporting the interpretability of ACT is valuable. In the revised manuscript, we therefore added a short paragraph in Section 3.2 explaining how ACT’s structure encourages simple, focused questions, and another in Section 4.1 highlighting the quantitative indicators (average question length and average path length, with detailed results in Appendix F) supporting this behavior. Below, we summarize the main elements of this evidence:
>
> 1. Across all experiments (beyond the ablation study), we use decision trees of depth 2–4. Each prediction therefore requires at most four yes/no questions, which we consider easily understandable for humans. In addition, we explicitly limit the complexity of each node’s question to at most two logical connectors (AND or OR)—as specified in the LLM instructions and detailed in Appendix A.2—so that the questions remain simple and interpretable.
> 2. In the five tables below (now in Appendix F), we report the average question length (in characters) and the average path length (number of nodes traversed) for each ACT configuration and dataset, averaged over the four LLMs.
> 3. For the DIAGNO dataset, we compare ACT’s learned questions with those commonly used in the medical literature for diagnosing tuberculosis. As shown in Appendix B.2, ACT identifies symptoms that closely match expert-validated diagnostic criteria, with substantial overlap between ACT’s questions and those reported by major medical sources.
>
> As shown in the tables below, average question length ranges from 78 to 155 characters, which we consider reasonable for interpretability. For reference, the generic seed question "Based on the provided example, does it belong to the positive
> class? (yes/no)" has a length of 87 characters. The average path length shows that even depth-4 trees typically require only three nodes. We also observe a trade-off: datasets with longer questions (JAILBREAK and SPAM) exhibit shorter paths, indicating that the model can consolidate multiple relevant features into a single, more informative question. Overall, each prediction relies on the yes/no answers to only about three concise and interpretable questions.
>
> ### DIAGNO
> | ACT (d,k) | Avg. Prompt length | Avg. Path length |
> |-----------|--------------------|------------------|
> | 3,10      | 92.00              | 2.35             |
> | 3,20      | 93.15              | 2.51             |
> | 4,10      | 78.05              | 3.03             |
> | 4,20      | 80.53              | 3.06             |
>
> ### SPAM
> | ACT (d,k) | Avg. Prompt length | Avg. Path length |
> |-----------|--------------------|------------------|
> | 2,5       | 118.00             | 1.33             |
> | 2,10      | 142.88             | 1.26             |
> | 3,5       | 94.18              | 1.36             |
> | 3,10      | 122.63             | 1.14             |
>
> ### JAILBREAK
> | ACT (d,k) | Avg. Prompt length | Avg. Path length |
> |-----------|--------------------|------------------|
> | 3,10      | 136.00             | 1.98             |
> | 3,20      | 155.00             | 2.13             |
> | 4,10      | 142.00             | 2.41             |
> | 4,20      | 150.33             | 2.57             |
>
> ### BANKCHURN
> | ACT (d,k) | Avg. Prompt length | Avg. Path length |
> |-----------|--------------------|------------------|
> | 3,10      | 90.50              | 2.92             |
> | 3,20      | 89.90              | 2.93             |
> | 4,10      | 82.20              | 3.10             |
> | 4,20      | 89.77              | 2.96             |
>
> ### IMDB
> | ACT (d,k) | Avg. Prompt length | Avg. Path length |
> |-----------|--------------------|------------------|
> | 3,10      | 81.30              | 1.97             |
> | 3,20      | 95.80              | 2.28             |

---

> ### Author Response · Authors · 2025-11-22
>
> >  Moreover, The paper should report and control data and compute exposure across methods by detailing the number of tokens, examples, and LLM calls for ACT, DSPy, and TextGrad.
>
> **Computational indicators:**
> We thank the reviewer for highlighting this important aspect and fully agree that a detailed analysis of computational cost is necessary. We have now included a detailed computational cost analysis in Appendix C, and we additionally summarize these results in **the Computational Cost section of the general response**. To address the concern, Figure. 7 in Appendix C reports the average wall-clock time and token usage on DIAGNO using Gemma-3-4B for both training and inference. Overall, ACT shows comparable wall-clock time and token count to TextGrad and DSPy at both training and inference, while providing interpretability and higher accuracy.
>
>
> >  Finally, report cost-normalized results, which can include tokens, runtime, and dollar cost per training run and per inference example—to clarify ACT’s efficiency and scalability.
>
> To answer, we now report token usage, monetary cost, test accuracy, and cost-normalized accuracy for each method on the DIAGNO dataset using the models of the GPT-4.1 family, as detailed in the newly added Appendix G. Figure 8 in Appendix G further shows that ACT’s test accuracy scales consistently with cost across model sizes, whereas TextGrad and DSPy do not exhibit comparable cost-dependent accuracy gains. The table below summarizes these metrics across the GPT-4.1 family, enabling a clear comparison of efficiency and scalability.
>
>
> ### Cost normalization and model scaling across the GPT-4.1 family on the DIAGNO dataset
> | Methods  | Model   | Cost  | Test Accuracy | Total Tokens (x100) | Cost Norm Acc |
> |----------|---------|-------|----------------|---------------------|----------------|
> | 3, 10    | Nano    |  1.09 | 66.90          |     55.18           |    61.54       |
> | 3, 20    | Nano    | 2.02  | 67.20          | 101.37              | 33.31          |
> | DSPY     | Nano    | 1.85  | 68.70          | 152.58              | 37.23          |
> | TextGrad | Nano    | 0.80  | 64.50          | 48.06               | 80.93          |
> | 3, 10    | Mini    | 3.09  | 77.30          | 48.08               | 24.98          |
> | 3, 20    | Mini    |  5.58 | 76.20          | 87.58               | 13.65          |
> | DSPY     | Mini    | 7.92  | 64.50          | 161.28              | 8.15           |
> | TextGrad | Mini    | 2.30  | 65.80          | 36.37               | 28.60          |
> | 3, 10    | GPT-4.1 | 19.08 | 80.20          | 53.71               | 4.20           |
> | 3, 20    | GPT-4.1 | 36.11 | 81.80          | 102.46              | 2.26           |
> | DSPY     | GPT-4.1 | 37.10 | 68.20          | 153.89              | 1.84           |
> | TextGrad | GPT-4.1 | 17.76 | 64.80          | 49.08               | 3.65           |
>
> Regarding cost-normalized accuracy (higher is better), TextGrad performs best in the Nano and Mini settings. Our ACT configuration with depth (3,10) remains competitive in these smaller regimes, generally ranking just behind TextGrad and also performing strongly on GPT-4.1. The (3,20) configuration is less cost-efficient due to its higher token usage, although it still performs slightly better than DSPy on the Mini and GPT-4.1 settings (but not on Nano). Overall, on GPT-4.1, ACT (3,10) achieves the most favorable accuracy–cost balance among the evaluated methods.

---

### Official Review · Reviewer_LjPb · 2025-10-31

**Soundness:** 3
**Presentation:** 3
**Contribution:** 3
**Rating:** 4
**Confidence:** 3

**Summary:**

The paper proposes Agentic Classification Tree (ACT), a method to exploit the natural language reasoning abilities of LLMs to design and train more linguistically friendly decision tree classifiers in NLP domains. The method takes inspiration from algorithms like CART and progressively splits based on optimized rules over time to maximize accuracy before the final leaf nodes. Experiments show it is more accurate than baselines and TF-IDF using CART. The final decision tree is also argued to me more interpretable.

**Strengths:**

The general idea of the paper is nice and intuitive, having user friendly splits with rules is quite appealing.

The quantitative results of the paper also look quite good, I am genuinely surprised this approach would be better than normal CoT prompting, or few shot prompting.

**Weaknesses:**

The method appears to be quite computationally expensive, which may drastically limit its practical usefulness.

There is nothing to say that the method will produce interpretable rules for the splits as far as I understand. Or at the very least, the logic behind the final tree will not be readily accessible to users. For example, why split on rule x at layer y?

The experiments are lacking appropriate baselines in my mind. I would like to see a comparison against e.g. a BERT classifier, or a TF-IDF with more layers in the model, or XGBoost, to understand the performance drop with the proposed method. I am also not sure why the authors limited the CART baseline to 3 depth, since their method often uses more than this, it doesn’t seem a fair comparison.

There are some sweeping claims in the paper about decision tree interpretability being easily understood and regulatable (e.g. line 141). I would want to see these tempered as I am not familiar with any evidence that this is true. Besides, it obviously depends on the tree depth, and whether or not it is an ensemble etc. These claims need to be tempered and/or reworded, and accompanied by better citations.

**Minor**

Typos line 112 and 127

**Questions:**

Why did you limit the CART TF-IDF to 3 depth when your own method went to 4?

Can you include other baselines such as a simple text classifier?

Why did you use these datasets? There are other more standard text classification domains. I would like to see more such as e.g. IMDB to show more generality.

Did you ever consider simply using the LLM to label concepts in the text and then train a decision tree classifier on this? There is a large literature on this, and it strikes me as an obvious baseline.

---

> ### Author Response · Authors · 2025-11-22
>
> We thank the reviewer for their insightful comments which helped us improve the paper. We answer each specific concern and question below.
>
> > The method appears to be quite computationally expensive, which may drastically limit its practical usefulness.
>
>
> **Computational cost.**
> We thank the reviewer for highlighting this important aspect and fully agree that a detailed cost analysis is necessary. We now include Figure 7 in Appendix C, reporting wall-clock time and token usage for training and inference. The **Computational Cost Table in the general response** also provides average wall-clock time and token counts for both training and inference on DIAGNO using Gemma-3-4B, together with additional context to help interpret these results. Overall, ACT exhibits computational costs comparable to TextGrad and DSPy while offering higher accuracy and interpretability.
>
>
> > There is nothing to say that the method will produce interpretable rules for the splits as far as I understand. Or at the very least, the logic behind the final tree will not be readily accessible to users. For example, why split on rule x at layer y?
>
> **Response (interpretability / rules).**
> ACT follows the classic “divide-and-conquer’’ principle of decision trees: at each node, the model only needs to separate one subset into two simpler ones. This local scope naturally leads the LLM to generate short, focused yes/no questions capturing the simplest distinction between the child nodes. As shown in Appendix B2–B4 (qualitative analysis and full trees for DIAGNO, Jailbreak, and Spam), these generated questions are indeed concise, human-readable, and aligned with domain-relevant concepts.
> Additionally, we would like to refer the reviewer to our response to Q2 of reviewer W3X5, where we provide new quantitative evidence (avg. question length and avg. path length, now included in Appendix F) supporting ACT’s interpretability.
>
> We agree that the underlying intuition behind why the rules tend to remain simple should be made clearer. In the revised manuscript, we therefore added a short paragraph in Section 3.2 explaining how ACT’s structure encourages simple, focused questions, and another in Section 4.1 highlighting the quantitative indicators (avg. question length and avg. path length) supporting this behavior.
>
>
> > The experiments are lacking appropriate baselines in my mind. I would like to see a comparison against e.g. a BERT classifier, or a TF-IDF with more layers in the model, or XGBoost, ... -Why did you limit the CART TF-IDF to 3 depth when your own method went to 4? - Can you include other baselines such as a simple text classifier?
>
> **CART depth.** We initially reported TF–IDF + CART with depth 3 because, in our experiments with depths {2, 3, 4}, this setting yielded the best average test accuracy across datasets. For DIAGNO, deeper trees overfitted, reducing test accuracy despite higher training accuracy, while for SPAM and JAILBREAK, the gains at depth 4 were only modest. To address the reviewer’s concern, the updated Table 2 now reports the best TF–IDF + CART test accuracy across depths 2, 3, 4, and additionally 5 for each dataset. We also provide the full train and test accuracies for all TF–IDF + CART configurations tested on the three original and two new datasets (BANKCHURN, IMDB) below and included them as Table 4 in Appendix D.
>
> **Additional baselines.**
> Following the reviewer’s recommendation, we now include TF–IDF + XGBoost (with multiple depths: d and number of trees: nt) as well as fine-tuned BERT and RoBERTa classifiers. We initially omitted these models because our focus was on more interpretable or prompt-based approaches, rather than on fine-tuned black-box models. While they substantially outperform classical CART, ACT remains competitive on several datasets while providing fully interpretable decision paths. Moreover, ACT can be built on top of any LLM accessible via an API, making it easy to use and applicable when stronger closed-source frontier models become available.

---

> ### Author Response · Authors · 2025-11-22
>
> ## Table Additional Baselines (Table 4 Appendix D)
> | Dataset / Split | CART (d=2) | CART (d=3) | CART (d=4) | CART (d=5) | XGB (d=2, nt=50) | XGB (d=2, nt=200) | XGB (d=3, nt=50) | XGB (d=3, nt=200) | XGB (d=4, nt=50) | XGB (d=4, nt=200) |
> |-----------------|------------|------------|------------|------------|------------------|-------------------|------------------|-------------------|------------------|-------------------|
> | **DIAGNO – Train**     |      82.0      |     84.3       |     86.5       |     **88.8**       |              86.2   |         94.7           |        89.2          |        98.7           |     93.5             |           **99.8**        |
> | **DIAGNO – Test**      |      77.5      |      **78.8**     |      73.7      |     76.3       |            81.0     |        81.5            |       81.8           |         81.7          |     82.2             |        **83.0**          |
> | **SPAM – Train**       |     87.8     |   92.3       |    93.5      |     **94.7**      |         97.8         |          99.7         |      98.7            |          99.8         |         99.2         |      **100.0**             |
> | **SPAM – Test**        |     85.8     |    89.3      |  90.7   | **92.7** |        94.3          |         96.2          |       93.8           |           **96.3**        |        94.5          |        96.0           |
> | **JAILBREAK – Train**  |       88.3   |      91.1    |      93.1    |      **94.3**    |          95.6        |       97.9            |         96.5         |          98.5         |        97.1          |        **99.0**           |
> | **JAILBREAK – Test**   |      86.4    |      92.4    |  **92.8**    |    91.9      |        95.2          |        **96.8**         |         95.2         |       96.4            |        95.2          |        **96.8**       |
> | **BANKCHURN – Train**  |         59.7   |      63.0    |     63.7     |    **66.7**  |        71.5          |        84.5           |      79.3            |         90.0          |         84.2         |      **93.2**         |
> | **BANKCHURN – Test**   |      62.7    |  62.8    |   62.8  |   **63.2** |           53.2       |      **56.7**         |        54.3          |         53.7          |        54.8          |       55.3            |
> | **IMDB – Train**       |     68.2     |     71.0     |     71.8    |  **76.2**    |      83.5            |       97.3            |      89.0            |          99.3         |         94.7         |       **99.8**        |
> | **IMDB – Test**        |  66.8  |  **66.8**  |     66.5    |  **66.8**   |        72.5          |      **76.2**        |       73.3           |         74.8          |        74.8          |       75.5            |
>
> > (Q3) Why did you use these datasets? There are other more standard text classification domains. I would like to see more such as e.g. IMDB to show more generality.
>
> **Dataset choice and generality. (Q3)**
> Following the reviewer’s suggestion, we additionally include a balanced subset of IMDB to provide a standard sentiment-classification task while keeping the sample size aligned with the other datasets. We also report results on a serialized version of BANKCHURN, where tabular attributes are converted into textual descriptions (see general response for additional details), offering a complementary domain. The new results are provided in the updated Table 2, which is already incorporated into the revised manuscript. Additionally, we provide an expanded discussion of all five datasets, along with our rationale for their inclusion, in Appendix A.4 of the revised paper.
>
> > (Q4) Did you ever consider simply using the LLM to label concepts in the text...
>
> **LLM to label concepts in the text. (Q4)**
> To evaluate this idea, we implemented an LLM-based concept pipeline in which (i) an LLM proposes a set of salient concepts from the training data, (ii) each example is mapped to a concept-presence vector using the LLM, and (iii) a CART classifier is trained on these concept features. We include this baseline as 'Con. tag.' in the revised experiments.
> Empirically, this approach performs below ACT. Results are reported in the updated Table 2.

---

> ### Author Response · Authors · 2025-11-22
>
> > There are some sweeping claims in the paper about decision tree interpretability being easily understood and regulatable....
>
> **Interpretability of decision trees.**
>
> First, it is important to specify that we are talking about "simple" decision trees here, meaning that tree-ensemble models are excluded. With that being said, Decision Trees are indeed generally considered to be more easily understandable and regulatable than most other machine learning models.
>
> Some arguments to support this claim:
>
>  - XAI community: it is widely accepted in the XAI community that the tree structure enables easy interpretations by humans. See e.g. [1, 2].
>  - Some user studies have also been conducted to support this claim. See e.g. [3], which measures through a user study how understandable decision trees are compared to e.g. neural networks.
>  - On the governance and regulation side: decision trees, and interpretable models in general, are often preferred due to their inherent interpretability. See e.g. this report by NIST (U.S. National Institute of Standards and Technology)" [4], which explicitly recommends to use Decision Trees (among other inherently interpretable models)  because they "lend themselves to more thorough documentation, audit, and governance".
>
>
> Although this understandability depends, as pointed out by the reviewer, on several parameters such as the tree depth, they remain therefore a good solution for interpretable models, especially in the context of the paper (text classification). In addition, we explicitly constrain the generated questions to at most two logical clauses (AND/OR), as detailed in Appendix A.2, and we limit the tree depth to ensure that the resulting rules remain short and easy to understand. Lastly, ACT interpretability indicators (average path length and average question length) are now reported in Appendix F.
>
> We included the citations above in the revised version of the paper to support our claims, and added a paragraph in Section 3.2 (Proposed Approach: Agentic Classification Tree) to clarify how ACT is designed to remain interpretable. We also added a paragraph in Section 4.1 (Experimental Results and Discussion) pointing to the quantitative interpretability indicators we now report for ACT.
>
> [1]"Interpretable Machine Learning", Molnar 2019.
>
> [2] "Stop explaining black box machine learning models for high stakes decisions and use interpretable models instead", Rudin 2019.
>
> [3] "Assessing the Local Interpretability of Machine Learning Models", Slack et al. 2019.
>
> [4] "AI RMF PLAYBOOK", NIST. url: <https://airc.nist.gov/docs/AI_RMF_Playbook.pdf>
>
> Finally, we thank the reviewer for pointing out the typos; we corrected them in the revised version of the paper.

---

### Author Response · Authors · 2025-11-22
**General response (Updated Table 2 with additional results and computational cost analysis in Appendix C)**

**Introduction:** We thank all reviewers for their time and constructive feedback. In this general section, we provide some comments on the updated version of Table 2 in the paper, now including F1 scores, two new datasets (BANKCHURN [1] and IMDB [3]) and additional baselines (TF-IDF + XGBoost, finetuned BERT and RoBERTa classifier, concept-tagging models, and a rule-based system). We also appreciate the reviewers’ request for a computational cost analysis. We fully agree that this aspect is essential, and provide the computational cost table below (which we now include as Figure 7 in Appendix C), reporting training and inference costs for DSPy, TextGrad, and ACT.

### Comments for Table 2

BANKCHURN [1] is a newly added tabular dataset that we serialize into text following [2]. We include it because it relies less on broad world knowledge and more on learning dataset-specific relationships between attributes and the churn label. In tasks such as SPAM, baselines like DSPy and TextGrad begin from a task-specific prompt (e.g., “Is this spam?”), whereas ACT starts from a neutral placeholder question (“Does this example belong to the positive class? (yes/no)”), which can give the baselines an advantage when the task aligns with pretraining knowledge. For BANKCHURN, however, the notion of “churn” does not correspond to a stable natural-language concept, so the initial task prompt offers less guidance and these methods may find it harder to identify effective rules. ACT’s local yes/no splits tend to be better suited to extracting the underlying attribute interactions, which leads to stronger performance in this setting. We also add a balanced subset of IMDB [3] to broaden the evaluation and assess generalization.

For TF-IDF + CART, we fixed the depth to 3 in the original paper, as it yielded the best average performance across datasets for depths {2, 3, 4}. In the revised version, we additionally include this analysis in the newly added Appendix D. We now evaluate depths {2, 3, 4, 5} and report in the updated Table 2 the configuration with the highest test accuracy per dataset. For TF-IDF + XGBoost, we evaluate eight settings (depth $\in \{2, 3, 4, 5\} \times$ number of trees $\in$ {50, 200}) and likewise report the configuration achieving the best test accuracy per dataset.

Fine-tuned BERT and RoBERTa, as well as TF-IDF + XGBoost, generally achieve the highest raw accuracy across datasets, but these models do not provide transparent, human-readable decision rules. In contrast, ACT offers more transparent decisions while remaining competitive with LLM-based prompting methods. When comparing only to the four LLM-dependent baselines (CoT, DSPy, TextGrad, and Concept Tagging), ACT achieves the highest test accuracy in 17/20 LLM–dataset combinations, with DSPy or TextGrad slightly ahead in the remaining cases. For F1-score, ACT is best in 15/20.

A few Jailbreak accuracy values were adjusted because some models refused to answer prompts they considered unsafe, and earlier evaluation scripts treated these refusals inconsistently. The corrections do not affect conclusions, and are marked in italics.

---

> ### Author Response · Authors · 2025-11-22
> **General response  Part 2**
>
> ## Computational Cost Table
> | Method              | Time (min) | Total Tokens (×100k) |
> |---------------------|------------|----------------|
> | *Training*         |            |                |
> | DSPy (BFSR, 8 demos)|     42     |       91       |
> | TextGrad (ep=12)    |    122     |       27       |
> | ACT (d=3, k=10)     |     43     |       49       |
> | ACT (d=4, k=20)     |    146     |      122       |
> | *Inference*        |            |                |
> | DSPy (BFSR, 8 demos)|    4.30    |       9.55     |
> | TextGrad (ep=12)    |   3.50     |      2.18      |
> | ACT (d=3, k=10)     |   4.36     |       3.26     |
> | ACT (d=4, k=20)     |   5.28     |       4.17     |
>
> ### Comments for the Computational Cost Table
> **Setting:** The computational cost table above compares the training and inference costs of DSPy, TextGrad, and two ACT configurations. The total token count refers to input and output tokens. To better interpret the table we want to provide some more details about the methods used in the paper:
> * For DSPy, we followed the setup used in the TextGrad paper [4]: BFSR (Bootstrapped Few-Shot Random Search) with 8 demonstrations. The method generates a reasoning trace and answer for each training instance, filters for correctly solved examples, constructs 13 candidate few-shot prompts (each with up to 8 examples), evaluates them on the full training set, and selects the best-performing prompt.
> * For TextGrad, we used a more generous configuration than the original TextGrad paper [4]: batch size 50 and 12 epochs, each consisting of 3 optimization steps. This allows TextGrad to optimize the prompt 36 times, each time evaluating the candidate on a validation set and retaining the best one.
> * For ACT, we report the cost of two configurations:
> (1) depth 3 with 10 optimization steps per node (cheapest ACT), and
> (2) depth 4 with 20 optimization steps per node (most expensive ACT).
>
> For inference, we report wall-clock time and token usage on the full test set.  All measurements were performed with Gemma-3-4B-IT on a single H200 GPU with 8 threads. All training and inference cost results are averaged over 5 independent runs.
>
> **Results:**
> TextGrad exhibits high wall-clock time despite lower token usage mainly because it issues fewer but substantially longer prompts; DSPy, in contrast, relies on a larger number of medium-sized prompts. ACT issues many short prompts to compute gini scores and a smaller number of longer prompts for node optimization. The cheapest ACT setting is competitive with both baselines in overall computational cost. Increasing ACT depth raises training cost but yields higher accuracy while preserving interpretability. At inference, TextGrad is the most economical, but ACT remains within the same order of magnitude and offers higher accuracy with interpretable outputs. DSPy shows similar wall-clock time to ACT but requires more tokens, namely because the few-shot prompts tend to be substantial in size.
>
>
> [1] Gaurav Topre. Bank customer churn dataset, 2025. URL https://www.kaggle.com/datasets/gauravtopre/bank-customer-churn-dataset.
>
> [2] Arzaghi, M., Dehghanpour Farashah, A., Carichon, F., and Farnadi, G. (2025). Intrinsic Meets Extrinsic Fairness: Assessing the Downstream Impact of Bias Mitigation in Large Language Models. arXiv. https://doi.org/10.48550/arXiv.2509.16462
>
> [3] Stanford NLP, IMDB Dataset, Hugging Face, 2020. URL: https://huggingface.co/datasets/stanfordnlp/imdb
>
> [4] Yuksekgonul, M., Bianchi, F., Boen, J., Liu, S., Huang, Z., Guestrin, C., & Zou, J. (2024). Textgrad: Automatic" differentiation" via text. arXiv preprint arXiv:2406.07496.

---

### Meta-Review · Area_Chair_Hxq1 · 2026-01-14

**Summary:**

The paper proposes Agentic Classification Trees (ACT), a method for building interpretable decision trees on unstructured text by using LLMs to generate and optimize natural language questions at each split. Combining the transparency of decision trees with the reasoning capabilities of LLMs is quite intuitive. The approach is technically elegant, and the qualitative examples of the resulting trees (e.g., for spam or jailbreak detection) demonstrate clear, human-readable decision paths. The authors were also responsive during the rebuttal, adding requested baselines (BERT, XGBoost) and a computational cost analysis. Despite the authors' extensive rebuttal efforts, several fundamental concerns regarding the practical utility and experimental scope of the method remain:

1) performance vs. complexity trade-off: the rebuttal revealed that ACT often underperforms compared to standard fine-tuned discriminative models or ensemble methods on the tested datasets . Given that ACT is computationally expensive (requiring iterative LLM calls for node optimization) compared to these traditional methods, the justification for using it on simple classification tasks is weak if it yields lower accuracy.

 2) toy-ish datasets: main experiment are on small, balanced datasets (e.g., DIAGNO and SPAM have only ~600 training samples) which function more as "toy" tasks than realistic high-stakes applications . While the authors added BANKCHURN and IMDB during the rebuttal, the overall evaluation remains focused on relatively simple binary classification problems where the heavy machinery of agentic tree optimization may be overkill.

The paper presents an interesting idea, but the current execution does not convincingly demonstrate that ACT offers a practical advantage over established baselines for the tasks presented. The high computational cost and lower accuracy compared to fine-tuned models, combined with the limited scale of evaluation, weigh against acceptance.

**Reviewer Concerns:**

Concerns addressed: computational cost analysis; baselines TF-IDF + XGBoost, fine-tuned BERT, and RoBERTa; added two new datasets: model scaling; the interpretability of the generated rules.

Concerns not addressed: performance vs. complexity trade-off and toy-ish/simple datasets.

**Reviewer Scores:**

NA

---

### Decision · Program_Chairs · 2026-01-26

Reject